# When to Make and Break Commitments?

**Alihan Hüyük**
University of Cambridge
ah2075@cam.ac.uk

**Zhaozhi Qian**
University of Cambridge
zq224@cam.ac.uk

**Mihaela van der Schaar**
University of Cambridge
The Alan Turing Institute
mv472@cam.ac.uk

## Abstract

In many scenarios, decision-makers must commit to long-term actions until their resolution before receiving the payoff of said actions, and usually, staying committed to such actions incurs continual costs. For instance, in healthcare, a newly-discovered treatment cannot be marketed to patients until a clinical trial is conducted, which both requires time and is also costly. Of course in such scenarios, not all commitments eventually pay off. For instance, a clinical trial might end up failing to show efficacy. Given the time pressure created by the continual cost of keeping a commitment, we aim to answer: *When should a decision-maker break a commitment that is likely to fail—either to make an alternative commitment or to make no further commitments at all?* First, we formulate this question as a new type of optimal stopping/switching problem called the *optimal commitment problem* (OCP). Then, we theoretically analyze OCP, and based on the insight we gain, propose a practical algorithm for solving it. Finally, we empirically evaluate the performance of our algorithm in running clinical trials with subpopulation selection.

## 1 Introduction

In many real-world settings, decision-makers must commit to long-term actions and wait until their resolution before receiving the payoff of said actions. Meanwhile, staying committed to such actions incurs continual costs. For instance, in portfolio management, it might take time for an asset to develop additional value after an initial investment, and keeping capital tied up in an asset comes with an opportunity cost for the investor (Markowitz, 1959; Merton, 1969; Karatzas and Wang, 2020). In an energy network, turning power stations on and off is not an immediate action, hence a sudden increase in energy demand can only be met with a delay after putting more stations into operation, and keeping stations operational obviously consumes resources (Rafique and Jianhua, 2018; Olofsson et al., 2022). In healthcare, a newly-discovered treatment can only be marketed to patients once a successful clinical trial that targets the said treatment is conducted, which both requires time and is also costly (Kaitin, 2010; Umscheid et al., 2011).

Of course, not all commitments eventually pay off: An asset might end up losing value despite investments, energy demands might shift faster than a network can react to, and a clinical trial might fail to show efficacy for the targeted treatment. Given the time pressure created by the continual cost of keeping a commitment, our goal in this paper is to answer the question: *When should a decision-maker break a commitment—thereby avoiding future costs but also forfeiting any potential returns—either to make an alternative commitment instead or to make no further commitments at all?* Solving this problem optimally requires a careful balance between *exploration* and *exploitation*: The earlier a commitment that is bound to fail is broken, the more resources would be saved (cf. exploitation); but the longer one is kept, the more information is revealed regarding whether the commitment is actually failing or might still succeed (cf. exploration)—and in certain cases, also regarding the prospects of similar commitments one could make instead.

Related problems are mostly studied within the context of adaptive experimentation and sequential hypothesis testing (see Section 5). As such, we focus on adaptive experimentation as our main application as well. More specifically, we consider the problem of selecting the target population of an adaptive experiment. Suppose an experimenter, who is interested in proving the efficacy of a new treatment, starts running an initial experiment that targets a certain population of patients. Incidentally, the treatment being tested is effective only for a relatively narrow subpopulation of patients but not for the wider population as a whole. Hence, an experiment targeting the overall population, but not the subpopulation specifically, will most probably fail to prove efficacy and prevent the deployment of the treatment for the patients who would have actually benefited from it, not to mention waste

time and resources (Moineddin et al., 2008; Lipkovich et al., 2017; Chiu et al., 2018). Of course, the experimenter has no knowledge of this in advance but the initial experiment they have set up would slowly reveal more information regarding the effects of the treatment and the fact that the ongoing experiment is bound to fail. In that case, we want to be able to determine at what point the experimenter has enough information to justify *breaking their commitment* to the initial experiment that targets too wide of a population to be successful, in favor of *making a new commitment* to a follow-up experiment that focuses on a narrower subpopulation instead?

**Contributions**   Our contributions are threefold: First, we formulate the problem of making and breaking commitments in a timely manner as a new type of optimal stopping/switching problem called the *optimal commitment problem* (OCP) (Section 2). The defining feature of OCP is that rewards are received only when a known time point is reached but costs are incurred continually, requiring commitment to actions but with incentive to abandon those commitments. As we will show later, OCP cannot be easily solved via conventional reinforcement learning techniques due to its non-convex nature. Second, we theoretically analyze a simplified case of OCP to identify the characteristics of the optimal solution (Section 3), and based on the insights we gain, propose a practical algorithm for the more general case (Section 4). Third, we empirically evaluate the performance of our algorithm in running experiments with subpopulation selection (Section 6). Before we move on, it should be emphasized that, although we predominantly consider adaptive experimentation as our main application, our contributions remain generally applicable to portfolio management, energy systems, and any other decision-making scenarios that require commitments to long-term actions.

## 2   OPTIMAL COMMITMENT PROBLEM

We first introduce the problem of optimal commitment from the perspective of running *experiments*. As far as our formulation is concerned, experiments are conducted to confirm the efficacy of an intervention by observing the outcome of the said intervention for subjects belonging to a particular population. However, this experiment-focused perspective does not limit the applicability of OCP; we stress its generality later at the end of the section. We provide a glossary of terms and notation in Appendix K.

**Populations**   Let $\mathcal{X}$ be a discrete set of *atomic-populations* such that every subject is only the member of exactly one atomic-population $x \in \mathcal{X}$. Denote with $\eta_x \in [0, 1]$ the probability of a subject being from atomic-population $x$ (such that $\sum_{x \in \mathcal{X}} \eta_x = 1$), and with $\Omega_x$ the distribution of outcomes for atomic-population $x$ such that the mean outcome $\theta_x = \mathbb{E}_{y \sim \Omega_x}[y]$ is the effect of some intervention for atomic-population $x$. Now, wider populations can be constructed by combining various atomic-populations. Let any $X \subseteq \mathcal{X}$ represent the *population* of subjects who belong to either one of the atomic-populations $\{x \in X\}$. Then, the probability of a subject being from population $X$ can be written as $\eta_X = \sum_{x \in X} \eta_x$, the probability of a subject being from atomic-population $x$ conditioned on the fact that they are from population $X$ can be written as $\eta_{x|X} = \eta_x / \eta_X$, and the average effect for population $X$ can be written as $\bar{\theta}_X = \sum_{x \in X} \eta_{x|X} \theta_x$.

**Experiments**   An experiment is largely characterized by the population it targets, its sample horizon, and its success criterion. During an experiment that targets population $X$, at each time step $t \in \{1, 2, \ldots\}$ that the experiment continues, first a subject from some atomic-population $x_t$ within the targeted population $X$ arrives with probability $\eta_{x_t|X}$, and then the outcome $y_t \sim \Omega_{x_t}$ for that subject is observed. This process generates an online dataset $\mathcal{D}_t = \{x_{t'}, y_{t'}\}_{t'=1}^t$. The experiment terminates when a pre-specified *sample/time horizon* $\tau$ is reached. Once terminated, the experiment is declared a success if $\rho(\mathcal{D}_\tau) = 1$, where $\rho : (\mathcal{X} \times \mathbb{R})^\tau \to \{0, 1\}$ is the *success criterion*, and declared a failure otherwise. Formally, the tuple $\psi = (X, \tau, \rho)$ constitutes an *experiment design*.

**Meta-experimenter**   Suppose a meta-experimenter is given a set of viable experiment designs $\Psi$ and is tasked with running at least one successful experiment. Each experiment $\psi \in \Psi$ has an associated *cost* $C_\psi \in \mathbb{R}_+$, which the experiment incurs per time step that it continues, and an associated *reward* $R_\psi \in \mathbb{R}_+$, which the experiment provides only if it eventually succeeds. The meta-experimenter aims to maximize utility—that is the difference between any eventual reward received and the total costs incurred by running experiments. They first pick an initial experiment $\psi^1 \in \Psi$ and start conducting it, which generates an online dataset $\mathcal{D}_t^1$ as described earlier. Now at each time step $t$, they need to decide whether they should stay committed to their initial decision and wait until $\psi^1$ terminates, or stop $\psi^1$ early in favor of starting a new experiment $\psi^2$. They might decide on the latter to avoid unnecessary costs if $\mathcal{D}_t^1$ already indicates $\psi^1$ is unlikely to succeed. If at some point a secondary experiment $\psi^2$ is started, now the meta-experiment has a similar decision to make

regarding whether to stop $\psi^2$ early in favor of starting a new experiment $\psi^3 \in \Psi$. This process continues until either an experiment finally succeeds or the meta-experimenter decides not to conduct any further experiments; let the random variable $n \in \{1, 2, \ldots\}$ be such that $\psi^n$ is the last experiment. We denote with $\psi^i = (X^i, \tau^i, \rho^i)$ the $i$-th experiment conducted by the meta-experimenter, and with $T^i$ the number of time steps for which the $i$-th experiment is conducted either until it was stopped by the meta-experimenter or the time horizon $\tau^i$ was reached. Denote with $\pi(t, \psi^i, \bar{\mathcal{D}}_t^i)$ the *decision-making policy* of the meta-experimenter, where $t$ is the current time step of the latest experiment $\psi^i$ and $\bar{\mathcal{D}}_t^i = (\cup_{j=1}^{i-1} \mathcal{D}_{T^j}^j) \cup \mathcal{D}_t^i$ is an aggregate dataset. We write (i) $\pi(t, \psi^i, \bar{\mathcal{D}}_t^i) = \psi^i$ if the meta-experiment decides to keep conducting the current experiment $\psi^i$, (ii) $\pi(t, \psi^i, \bar{\mathcal{D}}_t^i) = \psi' \neq \psi^i$ if the meta-experimenter decides to stop experiment $\psi^i$ and start experiment $\psi'$ instead, and (iii) $\pi(t, \psi^i, \bar{\mathcal{D}}_t^i) = \varnothing$ if the meta-experimenter decides not to conduct any further experiments.

**Objective** Once all experimentation is concluded, the meta-experimenter achieves the total *utility*

$$G = R_{\psi^n} \cdot \mathbb{1}\{T^n = \tau^n\} \cdot \rho^n(\mathcal{D}_{\tau^n}^n) - \sum_{i=1}^n C_{\psi^i} \cdot T^i . \tag{1}$$

Then, the *optimal commitment problem* is to find the optimal policy $\pi^* = \operatorname{argmax}_\pi \mathbb{E}_\pi[G]$ that maximizes the expected utility given $\Psi$, $\{\eta_x\}$. $\{R_\psi, C_\psi\}$ without knowing mean outcomes $\{\theta_x\}$ or outcome distributions $\{\Omega_x\}$. It is called the optimal *commitment* problem because each experiment $\psi = (X, \tau, \rho)$ only provides a reward if the meta-experimenter commits to incurring its costs for at least $\tau$ time steps, and the meta-experimenter needs to decide which experiment in $\Psi$ is the better commitment—or if there is any experiment worth committing to at all—adaptively.

**General applicability of OCP** Although we have described OCP from the perspective of (meta-)experiment design, it can potentially be useful in modeling many other problems as we have stressed during the introduction (see Table 1). For instance, in portfolio management, *atomic-populations* can be regarded as various *assets* one can invest in, then a *population* would correspond to a *portfolio*

Table 1: **Equivalent concepts across different domains.** OCP can model scenarios other than adaptive experimentation.

| Domain | Equivalent Concepts | |
|---|---|---|
| Adaptive experimentation | Atomic-population | Population |
| Portfolio management | Financial asset | Portfolio of assets |
| Energy systems | Power station | Network of stations |

*of assets*. Similar to experiments, when these portfolios require a time commitment (cf. $\tau$) before they provide their payoff (cf. $R_\psi$) and incur an opportunity cost (cf. $C_\psi$) in the mean time, the decision-making problem of managing when and which portfolio to invest in constitutes an instance of the optimal commitment problem. Another good examples is energy management, where *power stations* and the *networks* they form are akin to *atomic-populations* and *populations*. Since power stations cannot be turned on and off immediately, putting one in operation requires a certain amount of commitment.

## 3 WARM-UP: WHEN TO BREAK A SINGLE COMMITMENT?

In this section, to gather insights, we commence by analyzing a simplified instance of OCP. Later, in Section 4, using these insights, we construct a practical algorithm for solving a more general case of OCP. As the simplified instance, we only consider one atomic-population such that $\mathcal{X} = \{\mathcal{X}_0\}$ and one experiment design that targets this atomic-population such that $\Psi = \{\Psi_0 = (\mathcal{X}_0, \tau, \rho)\}$. Moreover, we assume that the outcomes are distributed normally with unit variance such that $\Omega \doteq \Omega_{\mathcal{X}_0} \doteq \mathcal{N}(\theta \doteq \theta_{\mathcal{X}_0}, 1)$ and the success criterion is a simple Z-test to see whether $\theta > 0$ such that $\rho(\mathcal{D}_\tau) \doteq \rho(\mu_\tau) \doteq \mathbb{1}\{\mu_\tau > \alpha/\sqrt{\tau}\}$, where $\mu_t = \sum_{(x_{t'}, y_{t'}) \in \mathcal{D}_t} y_{t'}/|\mathcal{D}_t|$ is the empirical mean outcome given dataset $\mathcal{D}_t$, and $\alpha$ determines the significance threshold for the test. Since there is just one viable experiment in this setting, the only decision that needs to be made at each time step is whether to keep conducting experiment $\psi^1 = \Psi_0$ or to stop all experimentation. For this decision to be interesting, we will also assume that $C \doteq C_{\Psi_0} > 0$—so that never stopping is not necessarily optimal—and $R \doteq R_{\Psi_0} > \tau C$—so that always stopping is not necessarily optimal either.

**Value and Q-functions** Since $t$ and $\mu_t$ are sufficient statistics to estimate the success probability of the experiment, it is also sufficient to only consider policies of the form $\pi(t, \mu)$. For a given policy $\pi$,

$$V^\pi(t, \mu) = \mathbb{E}[R \cdot \mathbb{1}\{T_t^\pi > \tau\} \cdot \rho(\mu_\tau) - C \cdot (\min\{T_t^\pi, \tau\} - t) \,|\, \mu_t = \mu] \tag{2}$$

$$Q^\pi(t, \mu) = \mathbb{E}[R \cdot \mathbb{1}\{T_{t+1}^\pi > \tau\} \cdot \rho(\mu_\tau) - C \cdot (\min\{T_{t+1}^\pi, \tau\} - t) \,|\, \mu_t = \mu] \tag{3}$$

are the value function, and the Q-function of conducting the experiment for at least one more time step respectively, where $T_t^\pi = \min\{t' \geq t : \pi(t', \mu_{t'}) = \varnothing\}$ is the first time step at or after time $t$ that policy $\pi$ decides to stop; let $V^* = V^{\pi^*}$ and $Q^* = Q^{\pi^*}$ be the optimal value and Q-functions. Note that the Q-factor of stopping all experimentation is always equal to zero for all policies. Hence, the optimal policy must be such that $\pi^*(t, \mu) = \Psi_0$ if $Q^*(t, \mu) > 0$ and $\pi^*(t, \mu) = \varnothing$ otherwise.

Once we identify the value and Q-functions, a naive attempt at finding the optimal policy would be to compute $V^*$ and $Q^*$ via dynamic programming as they would satisfy the following Bellman optimality conditions:

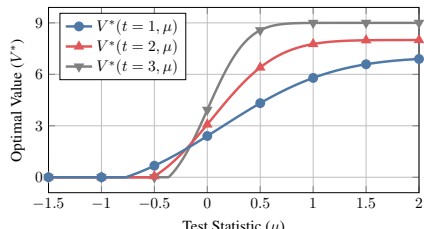

$$Q^*(t,\mu) = -C + \mathbb{E}[V^*(t+1,\mu_{t+1})|\mu_t = \mu] \quad (4)$$

$$V^*(t,\mu) = \max\{0, Q^*(t,\mu)\} \quad (5)$$

and $V^*(\tau,\mu) = R \cdot \rho(\mu)$. However, a major complication in applying dynamic programming methods to compute $V^*$ and $Q^*$ is that they are continuous functions in $\mu$. In the literature of partially-observable Markov decision processes (POMDPs), which OCP happens to be an instance

Figure 1: **Optimal value function $V^*(t,\mu)$ for $C = 1$, $R = 10$, $\tau = 4$, and $\alpha = 0$.** It can clearly be seen that neither $V^*$ nor $-V^*$ is convex in $\mu$ (cf. Proposition 1).

of (see Appendix A), the standard approach of addressing this complication would be to leverage the convexity of $V^*$ and $Q^*$, and approximate them with functions of the form $f(\mu) = \max_i a_i\mu + b_i$ (Spaan, 2012). However, this standard approach is not applicable in OCP because, in general, neither $V^*(t,\mu)$ nor $-V^*(t,\mu)$ is a convex function with respect to $\mu$ (see Figure 1):

**Proposition 1** (Non-convexity). *There exist a problem instance $(C,R,\tau,\alpha)$ and $t \in \{1,\ldots,\tau-1\}$ such that $\exists \mu,\mu' \in \mathbb{R}, p \in [0,1] : V^*(t,p\mu+(1-p)\mu') < pV^*(t,\mu)+(1-p)V^*(t,\mu')$ and $\exists \mu,\mu' \in \mathbb{R}, p \in [0,1] : -V^*(t,p\mu+(1-p)\mu') < -pV^*(t,\mu)-(1-p)V^*(t,\mu').*[1]

**Properties of the optimal policy**   Although identifying $\pi^*$ exactly by computing $V^*$ and $Q^*$ is challenging, we can still identify some properties that $\pi^*$ should have, which can then help us design a heuristic policy that we expect to perform well, albeit not optimally. First of all, the optimal policy $\pi^*$ should be a "thresholding-type" policy—that is the meta-experimenter should keep conducting the experiment as long as $\mu_t$ stays above a time-dependent threshold $\mu_t^*$ and should stop all experimentation the moment $\mu_t$ drops below that threshold (see the top panel of Figure 2):

**Proposition 2** (Thresholding). *For all problem instances $(C,R,\tau,\alpha)$, there exists time-dependent thresholds $\{\mu_t^* \in \mathbb{R}\}_{t=1}^{\tau-1}$ such that*

$$\pi^*(t,\mu) = \{\Psi_0 \quad if \, \mu > \mu_t^*; \quad \varnothing \quad otherwise\} \quad (6)$$

Intuitively, a higher test statistic $\mu_t$ means that the experiment is only more likely to succeed, hence if it is optimal to continue conducting the experiment when $\mu_t = \mu$, then it should also be optimal to continue when $\mu_t = \mu' > \mu$ (likewise, lower $\mu_t$ means success is even less likely hence $\pi^*(t,\mu) = \varnothing$ implies $\pi^*(t,\mu') = \varnothing$ for $\mu' < \mu$).

Moreover, the optimal policy $\pi^*$ must be "optimistic" that the experiment will succeed when making decisions. Consider a greedy policy $\pi^{\text{greedy}}$ that continues as long as the expected utility of committing fully to conducting the experiment until it terminates at $t = \tau$ is positive—that is $\pi^{\text{greedy}} = \Psi_0$ if and only if $V^{\pi^{(0)}}(t,\mu) > 0$ where $\pi^{(0)}$ is the policy that always waits until the experiment terminates such that $\pi^{(0)}(t,\mu) = \Psi_0$ for all $t,\mu$; $\pi^{\text{greedy}}$ is said to be greedy because the decision to continue is made assuming a full commitment to the experiment without considering the possibility to stop at a future time step. Then, whenever such greedy reasoning suggests continuing, the meta-experimenter should indeed continue. However, whenever the same reasoning suggests stopping, the meta-

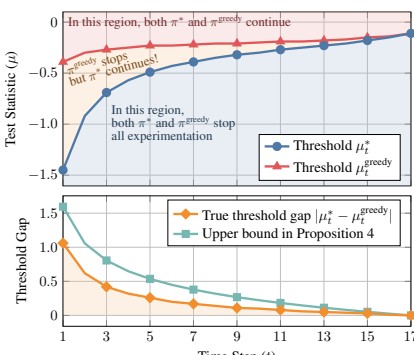

Figure 2: **Visualization of Propositions 2–4 for $C = 1$, $R = 50$, $\tau = 18$, and $\alpha = 0$.** Top figure shows how the thresholds in Proposition 2 evolve over time. Notice that $\mu_t^* \leq \mu_t^{\text{greedy}}$ hence why $\pi^*$ is optimistic (cf. Proposition 3). Bottom figure shows how the optimism of $\pi^*$ measured as $|\mu_t^* - \mu_t^{\text{greedy}}|$ decreases over time (cf. Proposition 4).

experimenter should be optimistic that the experiment will succeed and occasionally make the decision to continue instead—that is $\pi^*$ should be biased towards continuing (see the threshold gap in Figure 2):

**Proposition 3** (Optimism). *First, $\pi^{greedy}$ is also of thresholding type and there exists $\{\mu_t^{greedy} \in \mathbb{R}\}_{t=1}^{\tau-1}$ such that $\pi^{greedy}(t,\mu) = \Psi_0$ if and only if $\mu > \mu_t^{greedy}$. Moreover, for all $t \in \{1,\ldots,\tau-1\}$,*

$$\mu_t^* \leq \mu_t^{greedy} \quad \iff \quad \{\mu : \pi^*(t,\mu) = \Psi_0\} \supseteq \{\mu : \pi^{greedy}(t,\mu) = \Psi_0\} \quad (7)$$

---

[1]Proofs of all propositions are given in Appendix I.

Intuitively, the optimism of $\pi^*$ accounts for the information gained from observing more samples when the experiment is continued. Remember that $\pi^{\text{greedy}}$ estimates the reward to be received if the experiment is conducted until termination, and it stops whenever its estimate is negative. But, the estimate of $\pi^{\text{greedy}}$ has some uncertainty associated with it. Whenever it is uncertain enough that the reward to be received is actually negative; incurring the cost of continuing for one more time step, gaining new information, and forming a more certain estimate can lead to a more accurate decision and a higher overall utility. Finally, the optimism of $\pi^*$ has a strictly decreasing upper bound; denoting with $F(x) = (1/\sqrt{2\pi}) \int_{-\infty}^{x} e^{-(1/2)x^2}$ the c.d.f. of the standard normal distribution:

**Proposition 4** (Decreasing optimism). *For all $t \in \{1, \ldots, \tau - 1\}$,*

$$|\mu_t^* - \mu_t^{greedy}| \leq \sqrt{1/t - 1/\tau} \times \left( F^{-1}((\tau - t)C/R) - F^{-1}(C/R) \right) \quad (8)$$

Intuitively, as the experiment continues, the information gained from one individual sample decreases relative to the total information accumulated, hence the optimism of $\pi^*$ that accounts for the that information gain also decreases (see the bottom panel of Figure 2). Consider one extreme: When $t = \tau - 1$, there is no more information to be gained before the experiment terminates at $t = \tau$, hence $\pi^*$ should make the same decisions as $\pi^{\text{greedy}}$. Indeed, Proposition 4 implies that $\mu_{\tau-1}^* = \mu_{\tau-1}^{\text{greedy}}$.

## 4 A PRACTICAL ALGORITHM: BAYES-OCP

Summarizing our discussion in the previous section, we suspect the optimal policy to be (i) of thresholding type (cf. Proposition 2), (ii) optimistic (cf. Proposition 3), and (iii) increasingly more greedy (cf. Proposition 4). These findings are not a complete surprise as *optimism-in-the-face-of-uncertainty* is a well-known principle in solving online decision-making problems (Auer et al., 2002; Bubeck et al., 2012). Our earlier analysis shows rigorously that this principle holds for at least a special case of OCP and strengths our intuition that it should be applicable for more general cases of OCP as well.

Keeping properties (i–iii) in mind, we now propose a practical algorithm for solving OCP in a more general setting than the one we analyzed earlier. Let $|\mathcal{X}| \geq 1$ and $\Psi = \{(X, \tau, \rho) : X \in 2^{\mathcal{X}} \setminus \varnothing\}$ include all experiment designs that target a unique subpopulation within $\mathcal{X}$ for a given time horizon $\tau$ and success criterion $\rho$; let $C_X \doteq C_{(X,\tau,\rho)}$ and $R_X \doteq R_{(X,\tau,\rho)}$. We assume that the conditional power of performing a hypothesis test at time $\tau$ according to $\rho$—that is the probability of the test being successful conditioned on mean outcomes $\{\theta_x\}$—can be computed for interim datasets—that is

$$\mathcal{P}(X, \mathcal{D}_t; \{\theta_x\}) = \mathbb{E}_{x_{t'} \sim \{\eta_{x|X}\}_{x \in X}, y_{t'} \sim \mathcal{N}(\theta_{x_{t'}}, 1)}[\rho(\mathcal{D}_t \cup (\cup_{t'=t+1}^{\tau} \{x_{t'}, y_{t'}\}))] \quad (9)$$

can be evaluated efficiently. Then, based on this conditional power function, we define

$$\begin{aligned} &\mathcal{G}(X, \mathcal{D}_t; \{\theta_x\}) \\ &= R_X \cdot \mathcal{P}(X, \mathcal{D}_t) - C_X \cdot (\tau - t) \end{aligned} \quad (10)$$

as the expected utility of fully committing to an experiment and waiting until it terminates when the experiment targets population $X$, is currently at time step $t$, and has collected dataset $\mathcal{D}_t$ so far. Denote with $\mathcal{G}^{(0)}(X; \{\theta_x\}) = \mathcal{G}(X, \varnothing; \{\theta_x\})$ the same expected utility but for an experiment that is yet to start, and with $\mathcal{G}^{(0)}(\varnothing; \{\theta_x\}) = 0$ the utility of stopping all experimentation.

Our algorithm is called Bayes-OCP and is given in Algorithm 1. It maintains a posterior distribution $\mathcal{N}(\mu_x, \sigma_x^2)$ for each mean outcome $\theta_x$ assuming that, given mean $\theta_x$, outcomes are distributed normally with unit variance—that is $\Omega_x = \mathcal{N}(\theta_x, 1)$. These posteriors are only used in deciding which experiment to run next and *not* in determining whether the experiment was a success or not. Hence, even when the assumption of outcomes being normally distributed is violated, the integrity of the experiments would not be effected; only the performance of Bayes-

**Algorithm 1** Bayes-OCP

1: Initialize $\mu_x$ and $\sigma_x^2$ for all $x \in \mathcal{X}$
2: $X \leftarrow \mathcal{X}, \quad t \leftarrow 0, \quad \mathcal{D}_0 \leftarrow \varnothing$
3: Start experiment $\psi = (\mathcal{X}, \tau, \rho)$
4: **loop:**
5: $\quad t \leftarrow t + 1; \quad \mathcal{D}_t \leftarrow \mathcal{D}_{t-1} \cup \{x_t, y_t\}$
6: $\quad 1/\sigma_{x_t}^2 \leftarrow 1/\sigma_{x_t}^2 + 1$
7: $\quad \mu_{x_t} \leftarrow \mu_{x_t} + (y_t - \mu_{x_t})\sigma_{x_t}^2$
*(i) Identify a candidate subpopulation $X'$ to replace $X$:*
8: $\quad X' \leftarrow \varnothing$
9: $\quad$ **while** $X \setminus X' \supset \varnothing$:
10: $\quad\quad x^* \leftarrow \text{argmax}_{x \in X \setminus X'}$
$\quad\quad\quad \mathbb{E}_{\theta_x \sim \mathcal{N}(\mu_x, \sigma_x^2)}[\mathcal{G}^{(0)}(X' \cup \{x\}; \{\theta_x\})]$
11: $\quad\quad$ **if** $\mathbb{E}_{\theta_x \sim \mathcal{N}(\mu_x, \sigma_x^2)}[\mathcal{G}^{(0)}(X' \cup \{x^*\}; \{\theta_x\})]$
$\quad\quad\quad > \mathbb{E}_{\theta_x \sim \mathcal{N}(\mu_x, \sigma_x^2)}[\mathcal{G}^{(0)}(X'; \{\theta_x\})]$:
12: $\quad\quad\quad X' \leftarrow X' \cup \{x^*\}$
13: $\quad\quad$ **else: break**
*(ii) Decide whether to actually replace $X$ with $X'$:*
14: $\quad$ **if** $\mathbb{P}_{\theta_x \sim \mathcal{N}(\mu_x, \sigma_x^2)}\{\mathcal{G}^{(0)}(X'; \{\theta_x\})$
$\quad\quad\quad > \mathcal{G}(X, \mathcal{D}_t; \{\theta_x\})\} > \beta$:
15: $\quad\quad X \leftarrow X', \quad t \leftarrow 0, \quad \mathcal{D}_0 \leftarrow \varnothing$
16: $\quad\quad$ Start a new experiment $\psi = (X, \tau, \rho)$

OCP in managing various experiments would degrade (see Appendix C for related experiments). Making use of the posteriors it maintains, Bayes-OCP performs two steps at each iteration:

(i) *First,* a subpopulation $X' \subset X$ within the currently targeted population $X$ is identified as a potential candidate to target next; due to the combinatorial size of $\Psi$, it would not be practical to consider every subpopulation individually as a candidate for large $|\mathcal{X}|$. The ideal candidate would be the subpopulation with the largest expected utility: $X' = \operatorname{argmax}_{X' \subset X} \mathbb{E}_{\theta_x \sim \mathcal{N}(\mu_x, \sigma_x^2)}[\mathcal{G}^{(0)}(X'; \{\theta_x\})]$. But again due to the combinatorial size of the search space, Bayes-OCP employs a greedy algorithm instead and forms candidate subpopulations by combining, one by one, the atomic-subpopulations that increase the expected utility the most, until the expected utility no longer improves. Note that it is common to use greedy algorithms to solve combinatorial optimization problems (Lawler, 1976; Papadimitriou and Steiglitz, 1982).

(ii) *Then,* it is decided whether the current experiment targeting population $X$ should be stopped in favor of targeting candidate $X'$ identified earlier instead. A greedy strategy would have done so whenever $\mathbb{E}_{\theta_x \sim \mathcal{N}(\mu_x, \sigma_x^2)}[\mathcal{G}^{(0)}(X'; \{\theta_x\})] > \mathbb{E}_{\theta_x \sim \mathcal{N}(\mu_x, \sigma_x^2)}[\mathcal{G}(X, \mathcal{D}_t; \{\theta_x\})]$. But from our earlier analysis, we have learned that the optimal strategy is optimistic (cf. Proposition 3). As such, Bayes-OCP checks whether it is overwhelmingly likely that the alternative experiment has higher expected utility—that is whether $\mathbb{P}_{\theta_x \sim \mathcal{N}(\mu_x, \sigma_x^2)}\{\mathcal{G}^{(0)}(X'; \{\theta_x\}) > \mathcal{G}(X, \mathcal{D}_t; \{\theta_x\})\} > \beta$, where $\beta \in (1/2, 1)$ controls the decision-making threshold. When $\beta$ is large, we are more optimistic that the current experiment will succeed and require stronger evidence that the alternative experiment has higher expected utility. Note that, as the posteriors $\mathcal{N}(\mu_x, \sigma_x^2)$ get narrower, the optimism of this rule naturally decreases, which should be the case for the optimal strategy (cf. Proposition 4). As one extreme, the two switching rules become equivalent when $\{\sigma_x^2 \to 0\}$.

## 5 RELATED WORK

**Optimal stopping** Optimal commitment is essentially a new type of optimal stopping/switching problem. In typical optimal stopping problems (OSPs), the reward an agent can receive evolves based on a stochastic process and the goal of the agent is to determine the optimal time step to stop when the reward to be received is in some sense maximized (Shiryaev, 2007). Optimal commitment is unique in that a positive reward can only be received by *not* stopping until a pre-specified time horizon $\tau$. In optimal commitment, there is still a stochastic process (namely, samples $y_t$) that gradually reveals more information regarding what that positive reward will be at the end, however, the reward—or rather the cost—of stopping earlier is independent of this stochastic process (and is equal to $-tC$).

**Sequential hypothesis testing** Among other OSPs, optimal commitment is most closely related to sequential hypothesis testing (SHT), where an agent makes sequential observations regarding a given hypothesis and eventually needs to decide whether to reject the said (alternate) hypothesis or reject some null hypothesis (Wald and Wolfowitz, 1948; Yu et al., 2009; Drugowitsch et al., 2012; Shenoy and Angela, 2012; Zhang and Angela, 2013; Drugowitsch et al., 2014; Khalvati and Rao, 2015; Schönbrodt et al., 2017; Fauß et al., 2020). Rejecting the correct hypothesis provides a positive reward whereas waiting for more observations, while informative, is also costly as in OCP. It is well known that the optimal policy in the classic setting of SHT is a thresholding-type policy with fixed thresholds that do not vary over time: The null hypothesis is rejected if some test statistic gets above a threshold (and the alternate hypothesis is rejected if the same statistic gets below a different threshold).

Optimal commitment can be thought of as a SHT problem with the crucial difference that the meta-experimenter has *only* the option of discarding the alternate hypothesis (i.e. breaking a commitment), and once some time horizon is reached (i.e. when a commitment is kept), either the null hypothesis or the alternate hypothesis is automatically rejected according to some external success criterion $\rho$, regardless of what the meta-experimenters's decision might have been otherwise. As we have shown in Proposition 2, the optimal policy still remains a thresholding-type policy, but since there is now a deadline to discard the alternate hypothesis early, the thresholds become time-varying; in particular, they become less and less optimistic as the said deadline approaches (cf. Proposition 4).

Frazier and Angela (2007); Dayanik and Angela (2013); Alaa and van der Schaar (2016) consider SHT under stochastic deadlines, but different from optimal commitment, they still allow agents to reject both hypotheses at any time. In these works, the agent must make the rejection decision before the deadline is reached to be able to receive a positive reward, whereas in our case, the agent must wait until the deadline to see whether the null hypothesis will be rejected or not. Naghshvar and

Table 2: **Comparison of related experiment designs.** Optimal commitment is the only design that aims to decide both *when* an alternative population should be targeted—as opposed to switching the target population only at a fixed decision point—as well as *which* population to target among *many* potential candidates—as opposed to a simple binary decision of "overall population vs. sub-population" or "go vs. no-go".

| Design | Reference | *When?* | *Which?* |
|---|---|---|---|
| Randomized Controlled Trial (RCT) | Fisher (1935) | Never | Only the initial population |
| Adaptive Enrichment Design | Ondra et al. (2019) | Fixed decision point | Overall vs. fixed subpopulation |
| Adaptive Signature Design | Zhang et al. (2017) | Fixed decision point | Possibly any population |
| RCT with Futility Stopping | He et al. (2012) | Possibly any time | Go vs. no-go |
| Optimal Commitment | **(Ours)** | Possibly any time | Among multiple populations |

Javidi (2013); Jarrett and van der Schaar (2020) consider active versions of SHT where the agent is able to choose what type of observations to make. Our case is "passive" in the sense that the meta-experimenter cannot influence what kind of samples they are going to receive from the currently running experiment. Finally, optimal commitment, and SHT in general, can be thought of as more structured instances of partially-observed reinforcement learning (RL). As we have discussed earlier, the standard technique here relies on convex reward structures whereas the optimal value function in our case is not convex in general (cf. Proposition 1, see Appendix A for a detailed discussion).

**Adaptive experimentation** We introduced optimal commitment predominantly as a tool for population selection during an experiment. In clinical trials, dominant approach to population selection is adaptive enrichment (Mehta et al., 2009; Magnusson and Turnbull, 2013; Simon and Simon, 2013; Wang and Hung, 2013; Simon and Simon, 2018; Ondra et al., 2019; Thall, 2021) and adaptive signature designs (Freidlin and Simon, 2005; Freidlin et al., 2010; Mi, 2017; Zhang et al., 2017; Bhattacharyya and Rai, 2019). These designs are capable of adapting the target population of a trial as the trial continues, but unlike optimal commitment, they can only do so at fixed analysis points and not just at any time step. While adaptive signature designs can select arbitrary populations, adaptive enrichment designs are also limited by the number of pre-specified populations they can select between, which is typically only two: the overall population and an alternative subpopulation.

Optimal commitment is also related to clinical trial designs with futility stopping, where an experimenter might terminate a trial early once it becomes apparent that the said trial is highly unlikely to succeed (van der Tweel and van Noord, 2003; Lachin, 2005; He et al., 2012; Jitlal et al., 2012; Kimani et al., 2013; Chang et al., 2020). However, this does not consider the possibility of switching to a new trial that targets a different population. As we will see during our experiments, switching to an alternative experiment might prove preferable even before an ongoing experiment can be deemed futile. In such cases, optimal commitment can make more timely decisions. Table 2 summarizes the experiment designs related to optimal commitment. Finally, it is worth mentioning that there are several methods for managing clinical trials at a portfolio level—that is determining which clinical trial is to be conducted next (Rogers et al., 2002; Colvin and Maravelias, 2008; Graham et al., 2020). Trial management in this vain is orthogonal to optimal commitment: They are concerned with the success of multiple new treatments and make decisions on a trial-by-trial basis whereas we only ever consider a single intervention and make decisions regarding the target population on a sample-by-sample basis while experiments still continue. See Appendix H for extended related work.

## 6 EXPERIMENTS

We want to investigate how Bayes-OCP behaves in environments that differ in terms of ground-truth outcomes, for instance, what happens in environments where the original experiment is quite likely to succeed versus what happens in ones where switching to an alternative experiment is needed. To this end, we simulate experiments where mean outcomes are varied but other aspects of an experiment are fixed: In our environments, there are two atomic-populations, $\mathcal{X} = \{\mathcal{X}_A, \mathcal{X}_B\}$. Both atomic-populations have equal propensities $\eta_{\mathcal{X}_A} = \eta_{\mathcal{X}_B} = 1/2$ and the meta-experimenter has the same positively-biased prior for the mean outcome associated with each atomic-population: $\theta_{\mathcal{X}_A}, \theta_{\mathcal{X}_B} \sim \mathcal{N}(0.1, 0.1)$. Experiment designs targeting one or both of these atomic-populations all have the same time horizon $\tau = 600$ and success criterion $\rho(\mathcal{D}_\tau) = \mathbb{1}\{\sum_{(x_t, y_t) \in \mathcal{D}_\tau} y_t / |\mathcal{D}_\tau| > \alpha/\sqrt{\tau}\}$, where $\alpha = F^{-1}(95\%)$. So, experiments are powered to detect a positive mean outcome of 0.1 with probability $\sim 80$. Rewards are given by $R_X = 1000\eta_X^{0.1}$—the wider the target population is, the more people a successful intervention can be marketed to—and costs are given by $C_X = 1/\eta_X^{0.1}$—the narrower the target population is, the harder it becomes to find subjects eligible to participate.

**Benchmarks** We consider the meta-experiment designs summarized in Table 2 as benchmarks (see Appendix A.1 for an RL-based benchmark). *Conventional RCT* always targets the overall population and never stops early—that is it always conducts the experiment $\psi = (\{\mathcal{X}_A, \mathcal{X}_B\}, \tau, \rho)$ until its completion. *Adaptive Enrichment* performs an intermediary analysis at $t = \tau/2 = 300$ and greedily selects the experiment with the highest expected utility from $\Psi = \{(X, \tau, \rho)\}_{X \subseteq \{\mathcal{X}_A, \mathcal{X}_B\}}$. *Futility Stopping* is implemented via Bayes-OCP by initializing the set of all experiments as a singleton $\Psi = \{\Psi_0 = (\{\mathcal{X}_A, \mathcal{X}_B\}, \tau, \rho)\}$. Intuitively, futility stopping only decides whether or not to stop the initial experiment that targets the overall population early. *Bayes-OCP* is initialized with $\beta = 0.80$ (see Appendix E for a sensitivity analysis). We also consider an abla-

Table 3: **Performance comparison in various environment instances.** Bayes-OCP has the highest expected utility—and a smaller FWER then conventional RCTs—when averaged over all environment instances. This is because Bayes-OCP is a balanced design whose structure does not favor certain environment instances over others. As an example, compare it with conventional RCTs: RCTs do not have an adaptive structure hence they favor green environments where it is not necessary to adapt the target population of the initial experiment. *Instances favored/addressed partially

| Algorithms: | | Oracle RCT | RCT | Adaptive Enrichment | Futility Stopping w/ Bayes-OCP | Greedy Bayes-OCP | Bayes-OCP |
|---|---|---|---|---|---|---|---|
| **Favored Instances:** | | N/A | Green | Green/Amber* | Green/Red | Amber*/Red | **Balanced (incl. Amber)** |
| **All Instances** (100%) | Utility | 260.4 | -39.4 (6.7) | 106.5 (6.9) | 150.0 (3.5) | 32.6 (3.1) | **171.8 (3.6)** |
| | FWER | 0.0% | 0.3% (0.1%) | 0.2% (0.1%) | 0.1% (0.1%) | 0.0% (0.0%) | 0.1% (0.1%) |
| | Switches | 0.5 | 0.0 (0.0) | 0.4 (0.0) | 0.5 (0.0) | 1.0 (0.0) | 0.6 (0.0) |
| | Success | 75.2% | 56.1% (0.7%) | 53.2% (0.8%) | 45.4% (1.3%) | 10.5% (0.8%) | 52.4% (1.2%) |
| | T-to-S | 600.0 | 600.0 (0.0) | 600.0 (0.0) | 600.0 (0.0) | 607.5 (1.9) | 615.1 (1.2) |
| | T-to-F | 35.6 | 600.0 (0.0) | 548.9 (16.1) | 57.6 (4.6) | 3.0 (0.5) | 70.8 (8.2) |
| **Green Instances** (47.3%) | Utility | 389.6 | **388.7 (3.9)** | 385.6 (3.7) | 337.7 (5.7) | 63.1 (3.5) | 343.4 (7.3) |
| | FWER | 0.0% | 0.0% (0.0%) | 0.0% (0.0%) | 0.0% (0.0%) | 0.0% (0.0%) | 0.0% (0.0%) |
| | Switches | 0.0 | 0.0 (0.0) | 0.0 (0.0) | 0.1 (0.0) | 0.9 (0.0) | 0.1 (0.0) |
| | Success | 99.0% | 98.9% (0.4%) | 97.4% (0.7%) | 86.0% (1.4%) | 18.8% (0.9%) | 88.2% (2.0%) |
| | T-to-S | 600.0 | 600.0 (0.0) | 600.0 (0.0) | 600.0 (0.0) | 605.8 (1.5) | 602.8 (0.4) |
| | T-to-F | 600.0 | 600.0 (0.0) | 759.4 (36.4) | 46.6 (7.6) | 2.5 (0.5) | 62.3 (14.3) |
| **Amber Instances** (29.4%) | Utility | 258.6 | -300.3 (19.8) | -17.6 (6.5) | -5.3 (5.4) | 11.6 (3.4) | **63.2 (5.6)** |
| | FWER | 0.0% | 0.7% (0.3%) | 0.6% (0.3%) | 0.4% (0.3%) | 0.0% (0.0%) | 0.3% (0.2%) |
| | Switches | 1.0 | 0.0 (0.0) | 0.7 (0.0) | 0.8 (0.0) | 1.1 (0.0) | 0.9 (0.0) |
| | Success | 96.6% | 30.0% (2.0%) | 22.6% (1.5%) | 15.2% (2.0%) | 5.3% (1.0%) | 35.2% (1.8%) |
| | T-to-S | 600.0 | 600.0 (0.0) | 600.0 (0.0) | 600.0 (0.0) | 617.2 (5.9) | 663.9 (7.2) |
| | T-to-F | 600.0 | 600.0 (0.0) | 745.0 (13.1) | 78.3 (9.3) | 3.4 (0.6) | 104.4 (19.1) |
| **Red Instances** (23.3%) | Utility | 0.0 | -579.2 (4.1) | -304.2 (4.4) | -35.1 (1.7) | **-2.8 (1.1)** | -39.7 (3.4) |
| | FWER | 0.0% | 0.2% (0.3%) | 0.2% (0.3%) | 0.1% (0.2%) | 0.0% (0.0%) | 0.2% (0.3%) |
| | Switches | 1.0 | 0.0 (0.0) | 1.0 (0.0) | 1.0 (0.0) | 1.1 (0.0) | 1.0 (0.0) |
| | Success | 0.0% | 2.1% (0.4%) | 1.8% (0.5%) | 0.9% (0.3%) | 0.3% (0.4%) | 1.6% (0.7%) |
| | T-to-S | – | 600.0 (0.0) | 600.0 (0.0) | 600.0 (0.0) | 600.0 (0.0) | 634.8 (39.1) |
| | T-to-F | 0.0 | 600.0 (0.0) | 343.1 (8.1) | 38.9 (2.1) | 3.5 (1.4) | 45.8 (2.9) |

tion of Bayes-OCP where decisions are made greedily instead of optimistically (*Greedy Bayes-OCP*). As a baseline of maximum achievable performance, we consider an oracle (*Oracle RCT*) that always runs the RCT with the optimum target (or does not run any RCT at all if that happens to be optimal).

**Environments** A meta-experimenter's performance is specific to the environment instance. In particular, it depends on the ground-truth outcome distributions $\{\Omega_x\}$ for different populations. For example, an algorithm that always immediately stops the experiment would perform best when the mean outcome is negative. Hence, to faithfully evaluate the benchmarks, we need to focus on the *average* performance across different environments. To this end, we randomly generated 1000 environments (repeated five times to obtain error bars) with true mean outcomes $\theta_{\mathcal{X}_A}, \theta_{\mathcal{X}_B}$ sampled independently from $\mathcal{N}(0.1, 0.1)$. Given these means, outcome distributions are set to be Gaussian with unit variance such that $\Omega_x = \mathcal{N}(\theta_x, 1)$. Depending on the true mean outcome, these environments can be categorized into three groups: (i) *green instances* where the initial experiment targeting the overall population has the highest utility, (ii) *amber instances* where an alternative experiment that targets a subpopulation has the highest utility, and (iii) *red instances* where no experiment has positive utility hence running no experiments is the optimal decision.

Different benchmarks favor different instances (see the top row of Table 3): *Conventional RCTs* do not allow for any adaptation hence they favor green instances where the target population of the initial experiment does not need to be adapted. *Adaptive Enrichment* allows for adaptation but only at a certain time point, which is often too late to stop unsuccessful experiments (as in red instances). However, an adaptive enrichment design at least makes it possible to eventually target a subpopulation, even though it might be too late to do so at the pre-specified decision point, hence it partially accommodates amber instances. *Futility Stopping* decides between either continuing with the initial experiment or stopping all experimentation completely (targeting a subpopulation is not an option) hence it favors either green or red instances (but not amber instances). *Greedy Bayes-OCP* is pessimistic (or rather not optimistic enough) towards any ongoing experiment succeeding, hence it favors red instances where no experiment is likely to succeed. Similar to adaptive enrichment, Greedy Bayes-OCP at least allows subpopulations to be targeted hence it too partially accommodates amber instances.

**Main results** Performance of a meta-experimenter is primarily measured by Bayesian utility which is the expected utility averaged over randomly sampled environment instances (*Utility*). Remember that maximizing utility was our main objective, and as such, Bayes-OCP has the highest expected utility when averaged over all environment instances, see Table 3. Unlike other benchmarks, Bayes-OCP strikes a good balance in prioritizing all environment instances at the same time. This is because Bayes-

OCP (i) can make timely decisions—unlike Adaptive Enrichment—and (ii) is optimistic hence it does not stop likely-to-succeed experiments prematurely—unlike Greedy Bayes-OCP. More specifically,

(i) **Timeliness of Bayes-OCP:** Bayes-OCP has an advantage in amber and red instances over adaptive enrichment and futility stopping. Consider the example in Figure 3: While Bayes-OCP stops in a timely manner, adaptive enrichment can only stop at a fixed decision point and experiments with futility stopping only stop when the ongoing experiment is failing not as soon as a better alternative emerges. This underlines the *exploitative* aspect of Bayes-OCP—making and breaking commitments to maximize utility.

(ii) **Optimism of Bayes-OCP:** While a design that favors early stopping is obviously desirable in amber and red environments, how much it is favored should be moderated to also succeed in green environments. Consider the example in Figure 4: Greedy Bayes-OCP prematurely stops the initial experiment in a green environment while Bayes-OCP does not. Theoretically, we know that the optimal policy should be optimistic towards the ongoing experiment succeeding and be hesitant to stop to a certain extend. This underlines the *exploratory* aspect of Bayes-OCP—keeping a seemingly failing commitment still has value as it reveals more information regarding whether the commitment is actually failing.

In Table 3, in addition to *Utility*, we also report the family-wise error rate (*FWER*)—that is the frequency of runs where at least one experiment (denote it with $\psi^i$) is declared successful (i.e. $\rho^i(\mathcal{D}_\tau^i) = 1$) despite the mean outcome being negative for the targeted population (i.e. $\bar{\theta}_{X^i} < 0$)—the average number of times the target population has been switched (*Switches*), the probability of success which is defined as achieving positive utility (*Success*), the average time until a successful outcome (*Time-to-Success*, *T-to-S*), and the average time until until an unsuccessful outcome where all experimentation is stopped

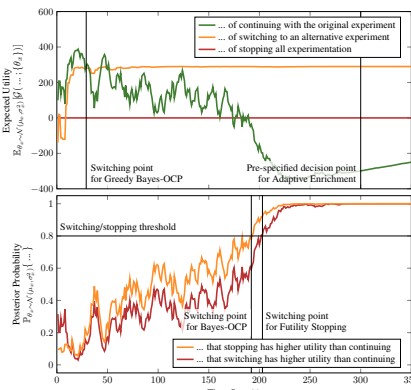

Figure 3: **Timeliness of Bayes-OCP.** Bayes-OCP is first to (correctly) stop the initial experiment in an amber instance (excluding Greedy Bayes-OCP). Adaptive enrichment can only stop at a pre-specified time, while futility stopping fails to consider switching to an alternative experiment, which is proven to be preferable earlier than stopping.

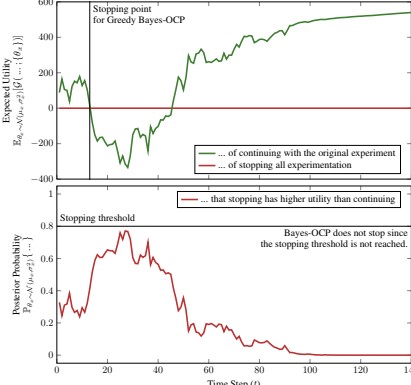

Figure 4: **Optimism of Bayes-OCP.** Greedy Bayes-OCP (incorrectly) stops due to initial noise in a green instance while Bayes-OCP does not stop since it is more optimistic (as the optimal policy should, cf. Proposition 3).

with negative utility (*Time-to-Failure*, *T-to-F*), see Appendix G for details. Importantly, Bayes-OCP does not compromise the error control of experiments, on the contrary, it even achieves a smaller FWER than conventional RCTs. This is because aggregate data is only ever used to select experiments, otherwise no two experiments consult each other's data when evaluating a success criterion so that the potential confoundedness that could have been caused by the adaptiveness of Bayes-OCP is avoided when declaring an experiment as successful (see Appendix B for a discussion on error control).

**Supplementary results** We also provide supplementary results: Appendix A.1 evaluates RL-based benchmarks, Appendix B.1 investigates error control, Appendix C considers environments with non-Gaussian outcomes, Appendix D considers environments with more than two atomic-populations, and Appendix E analyzes the sensitivity of Bayes-OCP's performance to its hyper-parameter $\beta$.

## 7 CONCLUSION

Two aspects of OCP require further discussion: (i) How can it be approached from the perspective of reinforcement learning? While OCP technically describes a special class of POMDPs, we have not found this to be constructive in finding a solution (see Appendix A). (ii) What are the implications of using Bayes-OCP in terms of error control? It has no impact on individual error rates and can be adapted to control FWER (see Appendix B). See Appendix F for a discussion on future work.

## ETHICS STATEMENT

As the main application of optimal commitment, we have focused on adaptive experimentation, particularly experiments that are run as part of clinical development. Clinical trials have a huge impact on the wellbeing of patients and this high-stakes nature of clinical trials naturally raises some ethical concerns; we discuss two major ones in this section. However before we start our discussion, it should be emphasized that clinical trials is not the only application domain of optimal commitment. As we have highlighted at the end of Section 2, our contributions are generally applicable to decision-making problems such as portfolio and energy management. Moreover, not all adaptive experiments are clinical and have the same high stakes as a clinical trial. For instance, A/B testing is common in online advertisement to determine what recommendation policies lead to more user engagement (Gui et al., 2015; Xu et al., 2015; Kohavi and Longbotham, 2017). Therefore, the ethical concerns we discuss here does not universally concern all possible applications of optimal commitment.

*The first concern* is how the designed error rate of an individual experiment is affected when multiple such experiments are managed together using Bayes-OCP in an adaptive manner, in particular, whether any error rate is inflated by the use of Bayes-OCP or not. We discuss error control in Appendix B with supplementary experiments. But briefly, Bayes-OCP has essentially no impact on the error rate of experiments on an individual level, and when controlling their family-wise error rate is also a concern, it can easily be adapted to accommodate this additional constraint as well.

*The second concern* is that an adaptive approach to population selection might lead to overly conservative experiments that unnecessarily limit the use of an effective treatment. As we have mentioned in the introduction to motivate the need for optimal commitment, when the treatment is effective only for a subpopulation (cf. amber instances in our experiments), population selection is absolutely necessary, otherwise the treatment is most likely to be found ineffective and discarded after an experiment that targets the overall patient population as a whole, which would deny the treatment for the subpopulation that would have benefited from it. On the flip side of this, when the treatment happens to be effective for everyone (cf. green instances in our experiment), population selection might lead to conducting a restrictive experiment that only targets a small subpopulation, which this time, would deny the treatment for the rest of the patient population. This is essentially the reason behind the performance drop between Bayes-OCP and conventional RCTs in green instances (see Table 3). There is a trade-off between the performance in amber instances and green instances; and Bayes-OCP achieves a better balance between the two compared with a conventional RCT as evidenced by its superior performance when *averaged* over all environment instances (again see Table 3); although it causes a drop in performance for green instances, it more than makes up for that drop in amber instances. This balance is partly controlled by how optimistic Bayes-OCP is, which is in turn dictated by its hyper-parameter $\beta$—larger $\beta$ leads to more optimistic decisions towards ongoing experiments, which favors green instances more then amber instances. We analyze the sensitivity of Bayes-OCP's performance to $\beta$ in Appendix E; and for all configurations that we have evaluated, Bayes-OCP always performs significantly better than a conventional RCT.

## REPRODUCIBILITY STATEMENT

All our experiments are based on synthetic simulations, hence our results can easily be reproduced by following the specifications in Section 6 without needing access to any private dataset. In order to aid reproducibility, we have rigorously described all our benchmarks in algorithmic form, similar to Algorithm 1, in Appendix J. Moreover, the source code necessary to reproduce our main results in Table 3 is made publicly available at `https://github.com/alihanhyk/optcommit` and `https://github.com/vanderschaarlab/optcommit`.

## ACKNOWLEDGMENTS

We would like to thank the reviewers and the members of the van der Schaar lab, for their valuable input, comments, and suggestions. This work was supported by the US Office of Naval Research (ONR) and the National Science Foundation (NSF, grant number 1722516).

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

## A  A REINFORCEMENT LEARNING PERSPECTIVE

Optimal commitment can be viewed as a partially-observed reinforcement learning problem. Let the tuple $(\mathcal{S}, \mathcal{A}, \mathcal{Z}, \mathcal{T}, \mathcal{O}, \mathcal{R})$ denote a partially-observable Markov decision process (POMDP), where $\mathcal{S}$ is the (unobserved) state space, $\mathcal{A}$ is the action space, $\mathcal{Z}$ is the observation space, $\mathcal{T} \in \Delta(\mathcal{S})^{\mathcal{S} \times \mathcal{S}}$ describes the transition dynamics, $\mathcal{O} \in \mathcal{Z}^{\mathcal{S}}$ describes the observation dynamics, and $\mathcal{R} \in \mathbb{R}^{\mathcal{S}}$ describes the reward dynamics. Then, OCPs as defined in Section 2 can also be expressed as a special class of POMDPs: Letting $\mathcal{Y} = \mathbb{R}$ denote the outcome space for clarity, $\mathfrak{D} = \cup_{t=0}(\mathcal{X} \times \mathcal{Y})^t$ be the space of all possible datasets $\mathcal{D}_t$, and $\mathfrak{O}$ be the space of all possible outcome distributions $\Omega$,

- $\mathcal{S} \doteq \{\varnothing\} \cup (\Psi \times \mathfrak{D} \times \mathfrak{O}^{\mathcal{X}})$, where states $s = (\psi, \mathcal{D}_t, \{\Omega_x\}_{x \in \mathcal{X}})$ consist of the ongoing experiment $\psi \in \Psi$, the dataset $\mathcal{D}_t \in \mathfrak{D}$ collected by the ongoing experiment so far, and the true outcome distributions $\{\Omega_x \in \mathfrak{O}\}$,

- $\mathcal{A} \doteq \{\varnothing\} \cup \Psi$,

- $\mathcal{Z} \doteq \{\varnothing\} \cup (\mathcal{X} \times \mathcal{Y})$,

- $\mathcal{T}(s = \varnothing, a) \doteq \varnothing$ and

$$\mathcal{T}(s = (\psi = (X, \tau, \rho), \mathcal{D}_t, \{\Omega_x\}), a)$$
$$\doteq \begin{cases} \varnothing & \text{if } a = \varnothing \\ \begin{aligned} s' &= (\psi, \mathcal{D}_{t+1} = \mathcal{D}_t \cup \{x_{t+1}, y_{t+1}\}, \{\Omega_x\}) \\ &\quad \text{s.t.} \quad x_{t+1} \sim \{\eta_{x|X}\}, y_{t+1} \sim \Omega_{x_{t+1}} \end{aligned} & \text{if } a = \psi \\ \begin{aligned} s' &= (\psi', \mathcal{D}_1 = \{x_1, y_1\}, \{\Omega_x\}) \\ &\quad \text{s.t.} \quad x_1 \sim \{\eta_{x|X'}\}, y_1 \sim \Omega_{x_1} \end{aligned} & \text{if } a = \psi' = (X', \tau', \rho') \neq \psi \,, \end{cases}$$

- $\mathcal{O}(s' = \varnothing) \doteq \varnothing$ and

$$\mathcal{O}(s' = (\psi, \mathcal{D}_{t+1} = \mathcal{D}_t \cup \{x_{t+1}, y_{t+1}\}, \{\Omega_x\})) \doteq (x_{t+1}, y_{t+1}) \,,$$

- $\mathcal{R}(s' = \varnothing) \doteq 0$ and

$$\mathcal{R}(s' = (\psi = (X, \tau, \rho), \mathcal{D}_{t+1}, \{\Omega_x\})) \doteq -C_\psi + R_\psi \cdot \mathbb{1}\{t+1 = \tau\} \cdot \rho(\mathcal{D}_{t+1}) \,.$$

Since ongoing experiments $\psi$ are completely dictated by actions, and datasets $\mathcal{D}_t$ collected by the ongoing experiments consist solely of observations $(x_t, y_t)$, the only unobserved component of the states in this POMDP is the true outcome distributions $\{\Omega_x\}_{x \in \mathcal{X}}$. Hence, the optimal policy should have the form $\pi(\psi, \mathcal{D}_t, b)$ where $b \in \Delta(\mathfrak{O}^{\mathcal{X}})$ denotes beliefs over $\{\Omega_x\}$—that is posterior distributions over the true outcome distributions. For instance, when $\Omega_x = \mathcal{N}(\theta_x, 1)$ as we have been assuming in Sections 3 and 4, posteriors over mean outcomes $\{\theta_x\}_{x \in \mathcal{X}}$, which are given by parameters $\{\mu_x, \sigma_x^2\}$ such that $\theta_x | \bar{\mathcal{D}}_t^i \sim \mathcal{N}(\mu_x, \sigma_x^2)$, constitute as beliefs.

Now although an OCP can be expressed as a POMDP, doing so is not particularly helpful in finding a solution. As we have already discussed in Section 3, the standard approach to solving a POMDP would be to use dynamic programming and compute the optimal value function $V^*$ and the optimal Q-function $Q^*$ iteratively according to Bellman optimality conditions

$$Q^*(b, a) = \mathbb{E}_{s \sim b, s' \sim \mathcal{T}(s,a), z' = \mathcal{O}(s'), b' | \{b, z'\}}[\mathcal{R}(s') + V^*(b')]$$
$$V^*(b) = \max_{a \in \mathcal{A}} Q^*(b, a) \,,$$

where $b' | \{b, z'\}$ denotes the updated belief $b'$ after having belief $b$ and making a new observation $z'$. When the state space $\mathcal{S}$ is discrete—or equivalently in our case, when the space of outcome distributions $\Omega \in \mathfrak{O}$ is discrete—$V^*$ and $Q^*$ happen to be convex functions, which makes it possible to perform these iterations efficiently by approximating $V^*$ and $Q^*$ using functions of the form $f(b) = \max\{a_i b + a_i'\}$ (Spaan, 2012). However, even in the simplest of cases where $\mathcal{S}$ is continuous—or equivalently, the space of outcome distributions $\Omega \in \mathfrak{O}$ is continuous, for instance when $\Omega_x = \mathcal{N}(\theta_x, 1)$—the convexity of $V^*$ and $Q^*$ no longer generally holds. In fact, we show in Proposition 1 that neither $V^*$ nor $-V^*$ is convex with respect to beliefs $b \equiv \{t, \mu\}$ for at least one instance of the simplified OCP that we have analyzed in Section 3.

Table 4: Performance comparison between Futility Stopping with RL-based algorithms and with Bayes-OCP.

| Algorithms: | | Oracle RCT | Futility Stopping w/ Discretized RL | Futility Stopping w/ Deep Q-learning | Futility Stopping w/ Bayes-OCP |
|---|---|---|---|---|---|
| **All Instances** (100%) | Utility | 260.4 | 131.8 (4.3) | 78.8 (3.1) | **150.0 (3.5)** |
| | FWER | 0.0% | 0.1% (0.1%) | 0.0% (0.0%) | 0.1% (0.1%) |
| | Switches | 0.5 | 0.6 (0.0) | 0.7 (0.0) | 0.5 (0.0) |
| | Success | 75.2% | 41.0% (1.0%) | 24.2% (0.8%) | 45.4% (1.3%) |
| | T-to-S | 600.0 | 600.0 (0.0) | 600.0 (0.0) | 600.0 (0.0) |
| | T-to-F | 35.6 | 54.3 (2.2) | 23.6 (1.8) | 57.6 (4.6) |
| **Green Instances** (47.3%) | Utility | 389.6 | 309.5 (4.1) | 185.0 (4.9) | **337.7 (5.7)** |
| | FWER | 0.0% | 0.0% (0.0%) | 0.0% (0.0%) | 0.0% (0.0%) |
| | Switches | 0.0 | 0.2 (0.0) | 0.5 (0.0) | 0.1 (0.0) |
| | Success | 99.0% | 80.9% (0.7%) | 47.7% (1.1%) | 86.0% (1.4%) |
| | T-to-S | 600.0 | 600.0 (0.0) | 600.0 (0.0) | 600.0 (0.0) |
| | T-to-F | 600.0 | 72.7 (7.1) | 11.0 (0.8) | 46.6 (7.6) |
| **Amber Instances** (29.4%) | Utility | 258.6 | -23.9 (5.4) | -16.6 (6.8) | **-5.3 (5.4)** |
| | FWER | 0.0% | 0.2% (0.2%) | 0.1% (0.1%) | 0.4% (0.3%) |
| | Switches | 1.0 | 0.9 (0.0) | 0.9 (0.0) | 0.8 (0.0) |
| | Success | 96.6% | 9.1% (0.8%) | 5.2% (1.3%) | 15.2% (2.0%) |
| | T-to-S | 600.0 | 600.0 (0.0) | 600.0 (0.0) | 600.0 (0.0) |
| | T-to-F | 600.0 | 66.1 (4.3) | 39.5 (4.4) | 78.3 (9.3) |
| **Red Instances** (23.3%) | Utility | 0.0 | -33.0 (1.6) | **-16.5 (5.2)** | -35.1 (1.7) |
| | FWER | 0.0% | 0.1% (0.2%) | 0.1% (0.2%) | 0.1% (0.2%) |
| | Switches | 1.0 | 1.0 (0.0) | 1.0 (0.0) | 1.0 (0.0) |
| | Success | 0.0% | 0.2% (0.2%) | 0.4% (0.4%) | 0.9% (0.3%) |
| | T-to-S | – | 600.0 (0.0) | 600.0 (0.0) | 600.0 (0.0) |
| | T-to-F | 0.0 | 33.7 (0.9) | 18.3 (4.8) | 38.9 (2.1) |

### A.1 EXPERIMENTS WITH REINFORCEMENT LEARNING BENCHMARKS

Having said all that, one naive way to still compute $V^*$ and $Q^*$ iteratively according to Bellman optimality conditions is to discretize the belief space. We call this benchmark *Discretized RL* and we use it to perform futility stopping—that is when $|\Psi| = 1$, deciding whether to stop the only viable experiment design early or not. Otherwise, the dimensionality of the belief state explodes combinatorially with respect to $|\Psi|$. We consider the same setting that we have considered during our experiments in Section 6 and compare the performance of *Futility Stopping with Discretized RL* with that of *Futility Stopping with Bayes-OCP*. When implementing discretized RL, instead of keeping track of the entire dataset $\mathcal{D}_t$, we only keep track of the sufficient statistic $\mu_t = \sum_{(x_{t'}, y_{t'})} y_{t'} / |\mathcal{D}_t|$, restrict the domain of $\mu_t$ to interval $[-0.3, 0.3]$, and discretize this interval into 100 equally-spaced bins. In addition to discretized RL, we also consider the approach proposed by Ni et al. (2022) for solving complex classes of POMDPs, which the optimal commitment problem is one of. Briefly, we employ deep Q-learning (as such, we call this benchmark *Deep Q-learning*) to train a neural network as an approximation of the Q-function $Q^*(b, a)$ using the POMDP we formalized earlier as a simulator. As the network architecture, we consider a multi-layer perceptron with two hidden layers of size 100 and with `tanh` activations.

Results are given in Table 4; futility stopping with Bayes-OCP performs better than futility stopping with discretized RL as well as futility stopping with deep Q-learning. In addition to the bad performance of discretized RL, it is also not feasible to scale it to use cases beyond futility stopping. When $|\Psi| > 1$, we would need to keep separate track of each $\mu_x$. Moreover, we would also need to start keeping track of the scale parameters $\{\sigma_x\}$ since it would now be possible to distribute samples among multiple atomic-populations in multiple ways by targeting different populations with different experiments (we no longer would be able to treat the target population of the only viable experiment design as the only atomic-population there is). Noting that $\sigma_x$'s already take discrete values with at least $\tau$-many possible values, merely increasing the number of viable experiments $|\Psi|$ from one to two causes the dimensionality of the belief space to jump from 100 to $\sim (100 \times 600)^2 = 36 \times 10^8$. Deep Q-learning performs even worse as it ignores all structure present in the optimal commitment problem, and instead, views the POMDP that describes it as a black-box simulator.

Table 5: Performance comparison of algorithms with family-wise error control.

| Algorithms: | | Oracle RCT | RCT | Adaptive Enrichment w/ Bonferroni Corr. | Futility Stopping w/ Bayes-OCP | Greedy Bayes-OCP w/ Bonferroni Corr. | **Bayes-OCP w/ Bonferroni Corr.** |
|---|---|---|---|---|---|---|---|
| **All Instances** (100%) | Utility | 260.4 | -39.4 (6.7) | 91.4 (5.4) | 150.0 (3.5) | 23.7 (2.2) | **158.7 (5.2)** |
| | FWER | 0.0% | 0.3% (0.1%) | 0.1% (0.1%) | 0.1% (0.1%) | 0.0% (0.0%) | 0.1% (0.1%) |
| | Switches | 0.5 | 0.0 (0.0) | 0.5 (0.0) | 0.5 (0.0) | 1.0 (0.0) | 0.6 (0.0) |
| | Success | 75.2% | 56.1% (0.7%) | 51.1% (0.7%) | 45.4% (1.3%) | 7.7% (0.5%) | 49.3% (1.5%) |
| | T-to-S | 600.0 | 600.0 (0.0) | 600.0 (0.0) | 600.0 (0.0) | 606.2 (2.3) | 616.6 (1.7) |
| | T-to-F | 35.6 | 600.0 (0.0) | 543.0 (15.7) | 57.6 (4.9) | 2.3 (0.3) | 65.8 (7.0) |
| **Green Instances** (47.3%) | Utility | 389.6 | **388.7 (3.9)** | 378.8 (3.1) | 337.7 (5.7) | 46.1 (3.3) | 325.8 (5.5) |
| | FWER | 0.0% | 0.0% (0.0%) | 0.0% (0.0%) | 0.0% (0.0%) | 0.0% (0.0%) | 0.0% (0.0%) |
| | Switches | 0.0 | 0.0 (0.0) | 0.0 (0.0) | 0.1 (0.0) | 1.0 (0.0) | 0.2 (0.0) |
| | Success | 99.0% | 98.9% (0.4%) | 96.1% (0.6%) | 86.0% (1.4%) | 13.8% (0.8%) | 84.7% (1.5%) |
| | T-to-S | 600.0 | 600.0 (0.0) | 600.0 (0.0) | 600.0 (0.0) | 604.9 (2.0) | 604.0 (0.7) |
| | T-to-F | 600.0 | 600.0 (0.0) | 768.6 (19.0) | 46.6 (7.6) | 2.0 (0.3) | 66.5 (17.7) |
| **Amber Instances** (29.4%) | Utility | 258.6 | -300.3 (19.8) | -51.9 (15.6) | -5.3 (5.4) | 8.3 (2.9) | **44.6 (4.6)** |
| | FWER | 0.0% | 0.7% (0.3%) | 0.3% (0.1%) | 0.4% (0.3%) | 0.0% (0.0%) | 0.2% (0.2%) |
| | Switches | 1.0 | 0.0 (0.0) | 0.8 (0.0) | 0.8 (0.0) | 1.1 (0.0) | 0.9 (0.0) |
| | Success | 96.6% | 30.0% (2.0%) | 18.2% (2.2%) | 15.2% (2.0%) | 3.8% (0.9%) | 30.7% (1.4%) |
| | T-to-S | 600.0 | 600.0 (0.0) | 600.0 (0.0) | 600.0 (0.0) | 613.4 (5.8) | 670.6 (8.9) |
| | T-to-F | 600.0 | 600.0 (0.0) | 724.6 (9.4) | 78.3 (9.3) | 2.6 (0.6) | 95.5 (18.4) |
| **Red Instances** (23.3%) | Utility | 0.0 | -579.2 (4.1) | -312.5 (2.3) | -35.1 (1.7) | **-2.2 (0.3)** | -37.0 (2.4) |
| | FWER | 0.0% | 0.2% (0.3%) | 0.2% (0.3%) | 0.1% (0.2%) | 0.0% (0.0%) | 0.1% (0.2%) |
| | Switches | 1.0 | 0.0 (0.0) | 1.0 (0.0) | 1.0 (0.0) | 1.1 (0.0) | 1.0 (0.0) |
| | Success | 0.0% | 2.1% (0.4%) | 1.1% (0.4%) | 0.9% (0.3%) | 0.1% (0.2%) | 1.1% (0.7%) |
| | T-to-S | – | 600.0 (0.0) | 600.0 (0.0) | 600.0 (0.0) | 600.0 (0.0) | 649.8 (62.6) |
| | T-to-F | 0.0 | 600.0 (0.0) | 334.9 (3.7) | 38.9 (2.1) | 2.4 (0.5) | 39.8 (2.5) |

# B DISCUSSION ON ERROR CONTROL

Bayes-OCP is a method for managing experiments—that is deciding what experiment to conduct and when—as opposed to a hypothesis testing strategy in and of itself. Implication of this in terms of error control is that the type 1 error of any individual experiment run by Bayes-OCP can always be controlled by choosing an appropriate experimental design, in particular, by specifying an appropriate success criterion $\rho$. This individual-level error control built into the design of each experiment is not compromised by Bayes-OCP; no aggregate data from multiple experiments is ever fed into the success criterion of one alone (see Section 2, experiment $\psi^i$ is successful if $\rho^i(\mathcal{D}_t^i) = 1$ not if $\rho^i(\bar{\mathcal{D}}_t^i) = 1$); and any assumptions made by Bayes-OCP regarding outcomes in Section 4, whether accurate or inaccurate, have no effect on the results produced by an external success criterion.

While Bayes-OCP does not compromise the individual error control of experiments, neither does it control their collective family-wise error rate (FWER)—that is the probability of at least one experiment among all that are conducted making a false discovery. Bayes-OCP views the problem of managing experiments purely as a utility maximization problem with no additional constraints. Within the scope of our discussion, the purpose of measuring FWER as a metric is to check empirically whether the individual error rates are inflated or not (note that FWER is a stricter notion of error than individual error rate). In practice, depending on how closely related the managed experiments are, controlling FWER might not necessarily be a concern. Let us highlight this: Any algorithm that manages experiments for long enough is bound to make at least one false discovery. Each year more than a thousand clinical trials are launched (that eventually post results) and more than half of these trials succeed (Takebe et al., 2018; Cli). If the type 1 error rate of all these trials were %5, we would expect at least 25 false discoveries in a year, which is more than one hence it would have put FWER of all real-world trials at almost $100\%$ when measured in a year-by-year basis. Of course, this is not problematic since not all clinical trials are related to each other closely enough to be considered a *family*.

## B.1 EXPERIMENTS WITH FAMILY-WISE ERROR CONTROL

When controlling FWER is of concern, Bayes-OCP can easily be adapted to satisfy this additional constraint by first limiting the number of total experiments that can be conducted—that is putting an upper bound on $n$—and then using well-established methods for family-wise error control such as Bonferroni correction or alpha spending functions (Demets and Lan, 1994) to adjust the success criteria of the viable experiments in $\Psi$. We run additional experiments to evaluate the performance of Bayes-OCP with Bonferroni correction. We consider the same setting that we have considered during our experiments in Section 6 except for one difference: We limit the number of experiments

Table 6: Performance comparison when the ground-truth outcome distributions are *not* Gaussian.

| **Algorithms:** | | Oracle RCT | RCT | Adaptive Enrichment | Futility Stopping w/ Bayes-OCP | Greedy Bayes-OCP | **Bayes-OCP** |
|---|---|---|---|---|---|---|---|
| **All Instances** (100%) | Utility | 266.5 | -38.4 (14.7) | 110.0 (9.5) | 150.8 (8.5) | 46.3 (3.8) | **178.2 (7.3)** |
| | FWER | 0.0% | 0.1% (0.1%) | 0.0% (0.1%) | 0.0% (0.1%) | 0.0% (0.0%) | 0.0% (0.1%) |
| | Switches | 0.5 | 0.0 (0.0) | 0.4 (0.0) | 0.5 (0.0) | 1.0 (0.0) | 0.6 (0.0) |
| | Success | 76.6% | 56.2% (1.5%) | 53.6% (1.5%) | 46.5% (1.8%) | 14.7% (1.1%) | 54.9% (1.7%) |
| | T-to-S | 600.0 | 600.0 (0.0) | 600.0 (0.0) | 600.0 (0.0) | 607.3 (1.0) | 617.2 (1.8) |
| | T-to-F | 32.5 | 600.0 (0.0) | 563.5 (9.9) | 65.8 (3.8) | 4.3 (0.5) | 81.9 (7.1) |
| **Green Instances** (48.0%) | Utility | 391.3 | **388.0 (4.1)** | 383.7 (3.1) | 343.3 (4.4) | 89.4 (6.7) | 348.7 (3.5) |
| | FWER | 0.0% | 0.0% (0.0%) | 0.0% (0.0%) | 0.0% (0.0%) | 0.0% (0.0%) | 0.0% (0.0%) |
| | Switches | 0.0 | 0.0 (0.0) | 0.0 (0.0) | 0.1 (0.0) | 0.9 (0.0) | 0.1 (0.0) |
| | Success | 99.1% | 98.8% (0.4%) | 97.3% (0.4%) | 87.6% (0.7%) | 26.6% (2.0%) | 89.3% (0.3%) |
| | T-to-S | 600.0 | 600.0 (0.0) | 600.0 (0.0) | 600.0 (0.0) | 605.8 (1.2) | 602.2 (1.3) |
| | T-to-F | 600.0 | 600.0 (0.0) | 710.1 (33.9) | 54.8 (10.0) | 4.2 (1.5) | 77.2 (10.6) |
| **Amber Instances** (29.9%) | Utility | 263.5 | -316.2 (18.2) | -13.1 (3.6) | -18.7 (8.8) | 14.1 (2.4) | **67.6 (6.4)** |
| | FWER | 0.0% | 0.2% (0.3%) | 0.1% (0.3%) | 0.1% (0.3%) | 0.0% (0.0%) | 0.1% (0.3%) |
| | Switches | 1.0 | 0.0 (0.0) | 0.7 (0.0) | 0.8 (0.0) | 1.1 (0.0) | 0.9 (0.0) |
| | Success | 97.2% | 28.4% (1.8%) | 22.6% (1.1%) | 14.7% (1.5%) | 6.3% (0.8%) | 39.3% (2.3%) |
| | T-to-S | 600.0 | 600.0 (0.0) | 600.0 (0.0) | 600.0 (0.0) | 617.5 (6.1) | 670.6 (6.1) |
| | T-to-F | 600.0 | 600.0 (0.0) | 765.5 (11.4) | 91.0 (6.0) | 4.3 (1.3) | 126.0 (16.6) |
| **Red Instances** (22.1%) | Utility | 0.0 | -588.3 (4.6) | -316.6 (12.2) | -37.0 (4.5) | **-3.8 (1.1)** | -41.7 (3.0) |
| | FWER | 0.0% | 0.0% (0.0%) | 0.0% (0.0%) | 0.0% (0.0%) | 0.0% (0.0%) | 0.0% (0.0%) |
| | Switches | 1.0 | 0.0 (0.0) | 1.0 (0.0) | 1.0 (0.0) | 1.1 (0.0) | 1.0 (0.0) |
| | Success | 0.0% | 1.2% (0.5%) | 1.0% (0.5%) | 0.5% (0.2%) | 0.2% (0.2%) | 1.6% (0.4%) |
| | T-to-S | – | 600.0 (0.0) | 600.0 (0.0) | 600.0 (0.0) | 600.0 (0.0) | 654.7 (21.8) |
| | T-to-F | 0.0 | 600.0 (0.0) | 341.9 (5.1) | 39.4 (4.2) | 4.2 (1.0) | 46.4 (3.9) |

that can be conducted by each algorithm as at most two, and we specify $\alpha = F^{-1}(0.975)$ for algorithms that can potentially run more than one experiment—namely, adaptive enrichment and (Greedy) Bayes-OCP—while we still specify $\alpha = F^{-1}(0.95)$ for algorithms that always run exactly one experiment—namely, RCT and futility stopping. These specifications ensure that FWER of all algorithms are bounded by %5. Results are given in Table 5; Bayes-OCP still performs the best when explicit control of FWER is required.

## C  EXPERIMENTS WITH MISSPECIFIED OUTCOME DISTRIBUTIONS

We consider the same setting that we have considered during our experiments in Section 6. Except now, the ground-truth outcome distributions are such that, when $y \sim \Omega_x$, $y = 1$ with probability $(\theta_x + 1)/2$ and $y = -1$ otherwise. In order to ensure that $\theta_x \in [-1, 1]$, we also sample ground-truth mean outcomes so that $\theta_x = 2p - 1$ where $p$ is distributed according to Beta distribution with $\alpha = 979/200$ and $\beta = 801/200$ (note that the mean and variance of $\theta_x$ remains the same as in our original experiments). Despite the fact that outcomes are now distributed in a non-Gaussian way, we leave the implementation of Bayes-OCP unchanged, which still assumes that outcomes distributions are Gaussian. So, there is now a mismatch between the structure of outcome distributions specified as part of Bayes-OCP and the ground-truth outcome distributions. Results are given in Table 6; Bayes-OCP still does not inflate FWER despite the misspecified outcome distributions.

## D  EXPERIMENTS WITH MORE ATOMIC-POPULATIONS

We repeat our main experiments with more than two atomic-populations, specifically we set $|\mathcal{X}| = 10$. As before, all atomic-populations have equal propensities such that $\eta_x = 1/10, \forall x \in \mathcal{X}$, and the meta-experimenter has the same positively-biased prior for the mean outcome associated with each atomic population: $\theta_x \sim \mathcal{N}(0.1, 0.1), \forall x \in \mathcal{X}$. We randomly generated 100 environment (repeated five times to obtain error bars), and the results are given in Table 7. We observe that Bayes-OCP still performs the best. These results confirm that a greedy approximation is suitable in identifying candidate experiments when the number of atomic-populations is large.

## E  SENSITIVITY ANALYSIS

Bayes-OCP has one hyper-parameter: $\beta$, which controls how optimistic the switching rule given in line 14 of Algorithm 1 is, from $\beta = 1/2$ meaning decisions are made greedily to $\beta = 1$ meaning

Table 7: Performance comparison when the number of atomic-populations is 10.

| Algorithms: | | RCT | Adaptive Enrichment | Futility Stopping w/ Bayes-OCP | Greedy Bayes-OCP | **Bayes-OCP** |
|---|---|---|---|---|---|---|
| **All** | Utility | 8.0 (39.2) | 143.9 (31.1) | 141.0 (27.5) | 40.3 (5.9) | **172.4 (23.8)** |
| **Instances** | FWER | 0.0% (0.0%) | 0.0% (0.0%) | 0.0% (0.0%) | 0.0% (0.0%) | 0.0% (0.0%) |
| **(100%)** | Switches | 0.0 (0.0) | 0.4 (0.0) | 0.5 (0.0) | 1.0 (0.0) | 0.6 (0.0) |
| | Success | 60.8% (3.9%) | 71.0% (3.6%) | 51.2% (4.7%) | 15.6% (2.6%) | 63.2% (4.2%) |
| | T-to-S | 600.0 (0.0) | 678.9 (8.3) | 600.0 (0.0) | 648.5 (13.2) | 647.5 (4.4) |
| | T-to-F | 600.0 (0.0) | 672.7 (58.8) | 130.4 (12.5) | 9.3 (5.0) | 200.5 (72.2) |

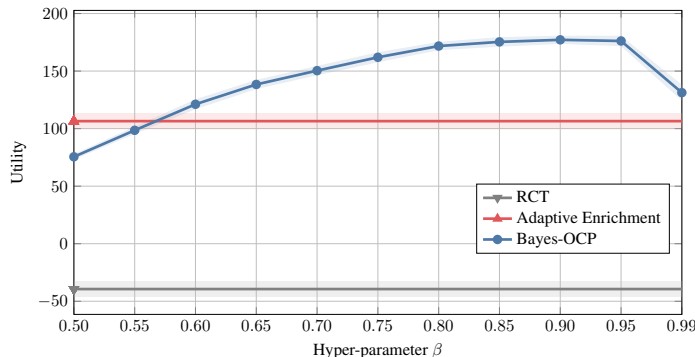

Figure 5: Utility achieved by Bayes-OCP for various values of hyper-parameter $\beta$.

decisions are so extremely optimistic that the original experiment will never be abandoned (as there will always be a chance that it succeeds). As with all online algorithms, tuning $\beta$ is challenging since no a priori data would be available to perform cross validation. However, a nice feature of Bayes-OCP is that $\beta$ is rather interpretable, it is the evidence required against the ongoing experiment: An alternative experiment is preferred over the ongoing experiment only if it is believed to be the better experiment with at least $\beta$-confidence. We evaluate the sensitivity of Bayes-OCP's performance to hyper-parameter $\beta$ in Figure 5; Bayes-OCP performs better than an RCT for all configurations and better than adaptive enrichment for most configurations.

## F FUTURE WORK

**Extending the scope of Bayes-OCP** One limitation of Bayes-OCP is that it only adapts the target population $X \subseteq \mathcal{X}$ of experiments but not the sample horizon $\tau$ or the success criterion $\rho$. We have chosen to focus on the selection of a target population since we believe the target population of an experiment to be the most critical design dimension to adjust adaptively. As we have already highlighted in our introduction, experiments with inflexible target populations can be problematic when responses to the treatment of interest are highly heterogeneous.

That being said, the high-level strategy of our proposed algorithm should still be applicable to adapting design dimensions other than the target population, namely $\tau$ and $\rho$. At a high level, Bayes-OCP first identifies a candidate experiment and then compares the identified experiment to the ongoing experiment in a n optimistic manner. Regardless of the given set of viable experiment design $\Psi$, one could still follow the same strategy; the only complication would be to adapt how candidate experiments are identified depending on what design dimension varies across experiment designs in $\Psi$.

For instance, when experiment designs varied in terms of $X$, a combinatorial search was required to identify good candidate experiments, for which we proposed a greedy strategy. When experiment designs vary in terms of $\rho$, a simple search over all possible $\rho$ would suffice for identifying candidate experiment. The case where experiment designs vary in terms of $\tau$ is more complex; optimal $\tau$ for an experiment would be dependent on unknown effects $\theta_x$; selecting a good candidate experiment would involve estimating the optimal $\tau$ given posteriors over $\theta_x$. This would be an interesting problem to explore as a future research direction.

**Performance guarantees**    While our theoretical results motivate the general use of an optimistic decision rule, they do not provide any guarantees about the performance of the specific rule we propose as part of Bayes-OCP. Another future research direction would be to prove an upper bound on the sub-optimality gap of Bayes-OCP.

## G    FURTHER DISCUSSION ON MAIN RESULTS

Table 3 report six metrics: Utility, FWER, Switches, Success, T-to-S, and T-to-F. We have already discussed the implications of Utility and FWER in Section 6. Here, we highlight other interesting phenomena regarding the remaining metrics. First, we see that Greedy Bayes-OCP switches experiments much more frequently compared with Bayes-OCP. This is because Greedy Bayes-OCP requires less evidence against the ongoing experiment when comparing it against an alternative experiment, whereas, Bayes-OCP favors the ongoing experiments more. Second, we see that a higher success probability does not necessarily also imply a higher utility. For instance, compare RCT with futility stopping, futility stopping is able to achieve higher utility than RCT by terminating risky experiments early and saving costs. However, this of course also means that futility stopping sees fewer experiments to completion hence leads to a lower success probability. Finally, we see that succeeding or failing early does not necessarily imply a higher utility either. Our best algorithm Bayes-OCP succeeds the latest on average as well as fails the latest compared with other benchmarks favoring red instances. This highlights the importance being conservative when making decisions, being optimistic, and favoring the status quo more than a potential adaptation.

## H    FURTHER DISCUSSION ON RELATED WORK

**Multi-armed bandits**    The optimal commitment problem is similar to a multi-armed bandit (MAB) problem (Auer et al., 2002; Bubeck et al., 2012) in some aspects: Like arms in a MAB problem, each experiment design $\psi$ has a random utility given by $R_\psi \cdot \rho(\mathcal{D}_\tau) - \tau C_\psi$, where $\mathcal{D}_\tau$ is the source of randomness, and the distribution of this utility is unknown. Also similar to a MAB problem, the overall goal is to sequentially select experiment designs (cf. arms) that yield the maximum cumulative utility. The main difference between the two problems is that, in a MAB problem, selecting an arm immediately reveals a sample from its random utility, while in optimal commitment, running an experiment $\psi$ just for one time step only incurs a cost of $C_\psi$; observing a full sample of its random utility requires the experiment to be run until its completion for $\tau$ consecutive time steps, without selecting any other experiment design in the meantime.

One can naively apply a MAB algorithm by viewing each viable experiment design as a unique arm, and by running experiments/arms selected by the algorithm until their completion to observe full samples from their unknown utility distributions. However, this obviously side steps the main question we want to answer in optimal commitment: When can we abandon a commitment—in this case, the decision to run an experiment/arm selection until its completion—before fully observing its outcome? Looking at optimal commitment from a MAB perspective reveals that there are two explore-exploit dilemmas present in optimal commitment: One is with respect to which experiment to select next, and the other is with respect to when to preemptively stop the current experiment (i.e. breaking a commitment). MAB algorithms address the former dilemma but not the latter.

**Task replication in parallel computing**    There is work (Ghare and Leutenegger, 2005; Wand et al., 2014; Wang et al., 2019) that focuses on the problem of when to kill existing tasks and relaunch them in parallel computing, which is related to optimal stopping/switching. However there, the focus is on reasoning about when a stochastic event (i.e. successful completion of a computational task) will occur *without any extra information* other than the fact that the event of interest has not occurred yet. In contrast, in our setting, the decision-maker needs to process a *streaming set of samples* to reason about the random outcome of an event that is scheduled to happen at a deterministic time point (here, the event is an experiment reaching its conclusion). This means that our problem has a completely different information structure when compared with the problem of task replication. More formally, we observe samples $y_t$ that are informative of whether $\rho(\mathcal{D}_\tau) = 1$ when $\tau$ is a fixed variable. In contrast, the problem of task replication would correspond to the setting where $\tau$ is a random variable with a known distribution and $\rho = 1$ always holds (hence no need to observe any samples $y_t$). Among optimal stopping/switching problems, the structure of our problem is more closely related to sequential hypothesis testing, which we have already covered in Section 5.

# I    PROOFS OF PROPOSITIONS

## I.1    PROOF OF PROPOSITION 1

We start by relating the optimal value function $V^*$ to the optimal Q-function $Q^*$. Letting $T_t^* = T_t^{\pi^*}$,

$$
\begin{aligned}
V^*&(t, \mu) \\
&= \mathbb{E}[R \cdot \mathbb{1}\{T_t^* > \tau\} \cdot \rho(\mu_\tau) - C \cdot (\min\{T_t^*, \tau\} - t)|\mu_t = \mu] \\
&= \mathbb{E}[\mathbb{1}\{\pi^*(t, \mu_t) = \varnothing\}(R \cdot \mathbb{1}\{T_t^* > \tau\} \cdot \rho(\mu_\tau) - C \cdot (\min\{T_t^*, \tau\} - t)) \\
&\qquad + \mathbb{1}\{\pi^*(t, \mu_t) = \Psi_0\}(R \cdot \mathbb{1}\{T_t^* > \tau\} \cdot \rho(\mu_\tau) - C \cdot (\min\{T_t^*, \tau\} - t))|\mu_t = \mu] \\
&= \mathbb{E}[\mathbb{1}\{\pi^*(t, \mu_t) = \varnothing\} \cdot 0 \\
&\qquad + \mathbb{1}\{\pi^*(t, \mu_t) = \Psi_0\}(R \cdot \mathbb{1}\{T_{t+1}^* > \tau\} \cdot \rho(\mu_\tau) - C \cdot (\min\{T_{t+1}^*, \tau\} - t))|\mu_t = \mu] && (11) \\
&= \mathbb{1}\{\pi^*(t, \mu) = \Psi_0\} \cdot \mathbb{E}[R \cdot \mathbb{1}\{T_{t+1}^* > \tau\} \cdot \rho(\mu_\tau) - C \cdot (\min\{T_{t+1}^*, \tau\} - t))|\mu_t = \mu] && (12) \\
&= \mathbb{1}\{Q^*(t, \mu) > 0\} \cdot Q^*(t, \mu) && (13) \\
&= \max\{0, Q^*(t, \mu)\}\,, && (14)
\end{aligned}
$$

where (11) holds since $\pi^*(t, \mu_t) = \varnothing \implies T_t^* = t$ and $\pi^*(t, \mu_t) = \Psi_0 \implies T_t^* \geq t+1 \implies T_t^* = \min\{t' \geq t : \pi^*(t', \mu_{t'}) = \varnothing\} = \min\{t' \geq t+1 : \pi^*(t', \mu_{t'}) = \varnothing\} = T_{t+1}^*$, (12) holds since $\mu_\tau \perp\!\!\!\perp \mathbb{1}\{\pi^*(t, \mu_t) = \Psi_0\}$ and $T_{t+1}^* \perp\!\!\!\perp \mathbb{1}\{\pi^*(t, \mu_t) = \Psi_0\}$ when conditioned on $\mu_t = \mu$, and (13) holds since $\pi^*(t, \mu) = \Psi_0 \iff Q^*(t, \mu) > 0$. Intuitively, the maximum possible value at a given time is achieved either by stopping immediately or by conducting the experiment for at least one more time step and then following the optimal policy thereafter.

Next, we observe that

$$
\begin{aligned}
\mathbb{P}\{\mu_{t+1} \leq \mu'|\mu_t = \mu\} &= \int \mathbb{P}\{\mu_{t+1} \leq \mu'|\theta, \mu_t = \mu\}d\mathbb{P}\{\theta|\mu_t = \mu\} \\
&= \int F\left(\mu' - \frac{\theta + t\mu}{t+1}; \frac{1}{(t+1)^2}\right)f(\theta - \mu; 1/t)d\theta \\
&= \iint \mathbb{1}\{\mu_{t+1} \leq \mu'\}f\left(\mu_{t+1} - \frac{\theta + t\mu}{t+1}; \frac{1}{(t+1)^2}\right)f(\theta - \mu; 1/t)d\mu_{t+1}d\theta \\
&= \iint \mathbb{1}\{\mu_{t+1} \leq \mu'\}f\left(\mu_{t+1} - \frac{y + (t+1)\mu}{t+1}; \frac{1}{(t+1)^2}\right)f(y; 1/t)d\mu_{t+1}dy \\
&= \iint \mathbb{1}\left\{x + \frac{y + (t+1)\mu}{t+1} \leq \mu'\right\}f(x; 1/(t+1)^2)f(y; 1/t)dxdy \\
&= \mathbb{P}_{\substack{X \sim \mathcal{N}(0, 1/(t+1)^2) \\ Y \sim \mathcal{N}(0, 1/t)}}\left\{X + \frac{Y}{t+1} \leq \mu' - \mu\right\} \\
&= \mathbb{P}_{X + Y/(t+1) \sim \mathcal{N}(0, 1/t - 1/t+1)}\left\{X + \frac{Y}{t+1} \leq \mu' - \mu\right\} \\
&= F(\mu' - \mu; 1/t - 1/t+1)\,, && (15)
\end{aligned}
$$

where $f(x; \sigma^2) = (1/\sqrt{2\pi\sigma^2})e^{-(1/2)x^2/\sigma^2}$ and $F(x; \sigma^2) = (1/\sqrt{2\pi\sigma^2})\int_{-\infty}^{x} e^{-(1/2)x'^2/\sigma^2}dx'$ are the p.d.f. and the c.d.f. of the Gaussian distribution with mean zero and variance $\sigma^2$ respectively. Hence $d\mathbb{P}\{\mu_{t+1} = \mu'|\mu_t = \mu\} = f(\mu' - \mu; 1/t - 1/t+1)d\mu'$.

Then, using the relationship between $V^*$ and $Q^*$ and the observation regarding $\mathbb{P}\{\mu_{t+1} \leq \mu'|\mu_t = \mu\}$, we drive the following Bellman optimality condition:

$$
\begin{aligned}
Q^*(t, \mu) &= \mathbb{E}[R \cdot \mathbb{1}\{T_{t+1}^* > \tau\} \cdot \rho(\mu_\tau) - C \cdot (\min\{T_{t+1}^*, \tau\} - t)|\mu_t = \mu] \\
&= -C + \mathbb{E}[R \cdot \mathbb{1}\{T_{t+1}^* > \tau\} \cdot \rho(\mu_\tau) - C \cdot (\min\{T_{t+1}^*, \tau\} - t - 1)|\mu_t = \mu] \\
&= -C + \int \mathbb{E}[R \cdot \mathbb{1}\{T_{t+1}^* > \tau\} \cdot \rho(\mu_\tau) - C \cdot (\min\{T_{t+1}^*, \tau\} - t - 1)|\mu_{t+1} = \mu'] \\
&\qquad\qquad\qquad\qquad\qquad\qquad\qquad\qquad\qquad \times d\mathbb{P}(\mu_{t+1} = \mu'|\mu_t = \mu) \\
&= -C + \int V^*(t+1, \mu')d\mathbb{P}(\mu_{t+1} = \mu'|\mu_t = \mu) \\
&= -C + \int V^*(t+1, \mu')f(\mu' - \mu; 1/t - 1/t+1)d\mu'
\end{aligned}
$$

$$= -C + \int V^*(t+1, \mu+z)f(z; {}^1\!/t - {}^1\!/{t+1})dz \qquad (16)$$

$$= -C + \int \max\{0, Q^*(t+1, \mu+z)\}f(z; {}^1\!/t - {}^1\!/{t+1})dz . \qquad (17)$$

For the problem setting where $C = 1$, $R = 2$, $\alpha = 0$, and $\tau = 2$, we have

$$
\begin{aligned}
V^*(1, \mu) &= \max\{0, -1 + \int V^*(2, \mu+z)f(z; {}^1\!/2)dz\} \\
&= \max\{0, -1 + 2\int \mathbb{1}\{\mu + z > 0\}f(z; {}^1\!/2)dz\} \\
&= \max\left\{0, -1 + 2\int_{-\mu}^{\infty} f(z; {}^1\!/2)dz\right\} \\
&= \max\{0, -1 + 2F(\mu; {}^1\!/2)\} \\
&= \begin{cases} 0 & \text{if } \mu < 0 \\ -1 + 2F(\mu; {}^1\!/2) & \text{if } \mu \geq 0 . \end{cases}
\end{aligned}
$$

Notice that, for $\mu > 0$,

$$
\begin{aligned}
\frac{d^2}{d\mu^2} V^*(1, \mu) &= \frac{d^2}{d\mu^2}\Big(-1 + 2F(\mu; {}^1\!/2)\Big) \\
&= \frac{d}{d\mu}\Big(2f(\mu; {}^1\!/2)\Big) \\
&= -(4/\pi)\mu e^{-\mu^2} < 0
\end{aligned}
$$

hence $V^*(1, \mu)$ is concave at least on interval $\mu \in (0, \infty)$ and is not a convex function. Moreover, $-V^*(1, \mu)$ cannot be a convex function—or equivalently $V^*(1, \mu)$ cannot be a purely concave function—either: For an arbitrary $\mu \in (0, \infty)$, $V^*(1, \mu) > 0$ and $V^*(1, -\mu) = 0$ hence $(1/2)V^*(1, \mu) + (1/2)V^*(1, -\mu) > 0$ but $V^*(1, (1/2)\mu + (1/2)(-\mu)) = V^*(1, 0) = 0$. $\qquad \square$

## I.2 PROOF OF PROPOSITION 2

We will prove the proposition by showing that

(i) $Q^*(t, \mu)$ is non-decreasing in $\mu$—that is $\mu < \mu' \implies Q^*(t, \mu) \leq Q^*(t, \mu')$,
(ii) $\lim_{t \to \infty} Q^*(t, \mu) = -(\tau - t)C + R > 0$, and
(iii) $\lim_{t \to -\infty} Q^*(t, \mu) = -C < 0$

for all $t \in \{1, \ldots, \tau - 1\}$ via mathematical induction. Notice that these three facts—together with the fact that $Q^*(t, \mu)$ is a continuous function in $\mu$ for $t \in \{1, \ldots, \tau - 1\}$—would imply the existence of a unique $\mu_t^*$ such that $Q^*(t, \mu_t^*) = 0$, $Q^*(t, \mu) > 0 \iff \mu > \mu_t^*$, and $Q^*(t, \mu) \leq 0 \iff \mu \leq \mu_t^*$, which in turn would imply that

$$
\pi^*(t, \mu) = \begin{cases} \Psi_0 & \text{if } \mu > \mu_t^* \iff Q^*(t, \mu) > 0 \\ \varnothing & \text{if } \mu \leq \mu_t^* \iff Q^*(t, \mu) \leq 0 , \end{cases}
$$

meaning the optimal policy $\pi^*$ is indeed of "thresholding -type" as the proposition states.

First, we observe the following base cases for $t = \tau - 1$:

(i) $Q^*(\tau - 1, \mu)$ is non-decreasing in $\mu$. When $\mu < \mu'$,

$$
\begin{aligned}
Q^*(\tau - 1, \mu) &= -C + \int V^*(\tau, \mu+z)f(z; {}^1\!/{(\tau-1)} - {}^1\!/\tau)dz \qquad (18) \\
&= -C + R\int \mathbb{1}\{\mu + z > \alpha/\sqrt{\tau}\}f(z; {}^1\!/{(\tau-1)} - {}^1\!/\tau)dz \\
&\leq -C + R\int \mathbb{1}\{\mu' + z > \alpha/\sqrt{\tau}\}f(z; {}^1\!/{(\tau-1)} - {}^1\!/\tau)dz \qquad (19) \\
&= Q^*(\tau - 1, \mu') ,
\end{aligned}
$$

where (18) is due to (16), and (19) holds since $\mu + z > \alpha/\sqrt{\tau} \implies \mu' + z > \mu + z > \alpha/\sqrt{\tau}$.

(ii) $\lim_{\mu \to \infty} Q^*(\tau - 1, \mu) = -C + R > 0$ since

$$
\lim_{\mu \to \infty} Q^*(\tau - 1, \mu) = \lim_{\mu \to \infty}\Big(-C + R\int \mathbb{1}\{\mu + z > \alpha/\sqrt{\tau}\}f(z; {}^1\!/{(\tau-1)} - {}^1\!/\tau)dz\Big)
$$

$$= \lim_{\mu \to \infty} \left( -C + R \int_{\alpha/\sqrt{\tau}-\mu}^{\infty} f(z; 1/(\tau-1) - 1/\tau)dz \right)$$

$$= -C + R \int f(z; 1/(\tau-1) - 1/\tau)dz$$

$$= -C + R .$$

(iii) $\lim_{\mu \to -\infty} Q^*(\tau-1, \mu) = -C < 0$ since

$$\lim_{\mu \to -\infty} Q^*(\tau-1, \mu) = \lim_{\mu \to -\infty} \left( -C + R \int \mathbb{1}\{\mu + z > \alpha/\sqrt{\tau}\} f(z; 1/(\tau-1) - 1/\tau)dz \right)$$

$$= \lim_{\mu \to -\infty} \left( -C + R \int_{\alpha/\sqrt{\tau}-\mu}^{\infty} f(z; 1/(\tau-1) - 1/\tau)dz \right)$$

$$= \lim_{\mu \to -\infty} \left( -C + R \left( 1 - \int_{-\infty}^{\alpha/\sqrt{\tau}-\mu} f(z; 1/(\tau-1) - 1/\tau)dz \right) \right)$$

$$= -C + R \left( 1 - \int f(z; 1/(\tau-1) - 1/\tau)dz \right)$$

$$= -C .$$

Then, we show that the following inductive cases hold for $t \in \{\tau-1, \ldots, 2\}$:

(i) Given that $Q^*(t, \mu)$ is non-decreasing in $\mu$, $Q^*(t-1, \mu)$ is also non-decreasing in $\mu$. Similar to the base case, when $\mu < \mu'$,

$$Q^*(t-1, \mu) = -C + \int \max\{0, Q^*(t, \mu+z)\} f(z, 1/(t-1) - 1/t)dz \qquad (20)$$

$$\leq -C + \int \max\{0, Q^*(t, \mu'+z)\} f(z, 1/(t-1) - 1/t)dz$$

$$= Q^*(t-1, \mu') ,$$

where (20) is due to (17).

(ii) Given $\lim_{\mu \to \infty} Q^*(t, \mu) = -(\tau-t)C + R$—and also given that $Q^*(t, \mu)$ is non-decreasing in $\mu$—we have $\lim_{\mu \to \infty} Q^*(t-1, \mu) = -(\tau-t+1)C + R > 0$, which can be shown using the sandwich theorem:

$$Q^*(t-1, \mu) = -C + \int \max\{0, Q^*(t, \mu+z)\} f(z, 1/(t-1) - 1/t)dz$$

$$\leq -C + \int \max\{0, \lim_{\mu' \to \infty} Q^*(t, \mu')\} f(z, 1/(t-1) - 1/t)dz$$

$$\leq -C + (-(\tau-t)C + R) \int f(z, 1/(t-1) - 1/t)dz$$

$$= -(\tau-t-1)C + R . \qquad (21)$$

$$Q^*(t-1, \mu) = -C + \int \max\{0, Q^*(t, \mu+z)\} f(z, 1/(t-1) - 1/t)dz$$

$$\geq -C + \int_{-|\mu|^{1/2}}^{\infty} \max\{0, Q^*(t, \mu+z)\} f(z, 1/(t-1) - 1/t)dz$$

$$\geq -C + \int_{-|\mu|^{1/2}}^{\infty} \max\{0, Q^*(t, \mu - |\mu|^{1/2})\} f(z, 1/(t-1) - 1/t)dz$$

$$\geq -C + Q^*(t, \mu - |\mu|^{1/2}) \int_{-|\mu|^{1/2}}^{\infty} f(z, 1/(t-1) - 1/t)dz , \qquad (22)$$

Finally, observing that

$$\lim_{\mu \to \infty} (22) = \lim_{\mu \to \infty} \left( -C + Q^*(t, \mu - |\mu|^{1/2}) \int_{-|\mu|^{1/2}}^{\infty} f(z, 1/(t-1) - 1/t)dz \right)$$

$$= -C + \left( \lim_{\mu' \to \infty} Q^*(t, \mu') \right) \int f(z, 1/(t-1) - 1/t)dz$$

$$= -(\tau-t-1)C + R ,$$

together with bounds (21) and (22), we obtain $\lim_{\mu \to \infty} Q^*(t-1, \mu) = -(\tau-t+1)C + R$.

(iii) Given $\lim_{\mu \to -\infty} Q^*(t, \mu) = -C < 0$—and also given that $Q^*(t, \mu)$ is non-decreasing in $\mu$ and $\lim_{\mu \to \infty} Q^*(t, \mu) > 0$ so that $\mu_t^*$ exists—we have $\lim_{\mu \to -\infty} Q^*(t-1, \mu) = -C < 0$, which again can be shown using the sandwich theorem:

$$Q^*(t-1, \mu) = -C + \int \max\{0, Q^*(t, \mu+z)\} f(z, 1/(t-1) - 1/t)dz$$

$$\geq -C \, . \tag{23}$$

$$Q^*(t-1, \mu) = -C + \int \max\{0, Q^*(t, \mu+z)\} f(z, 1/(t-1) - 1/t) dz$$

$$= \int_{\mu_t^* - \mu}^{\infty} Q^*(t, \mu+z) f(z, 1/(t-1) - 1/t) dz \tag{24}$$

$$\leq -C + R \int_{\mu_t^* - \mu}^{\infty} f(z, 1/(t-1) - 1/t) dz \tag{25}$$

$$\leq -C + R \left( 1 - \int_{-\infty}^{\mu_t^* - \mu} f(z, 1/(t-1) - 1/t) dz \right) , \tag{26}$$

where (24) holds since $Q^*(t, \mu+z) > 0$ if and only if $z > \mu_t^* - \mu$ and $\max\{0, Q^*(t, \mu+z)\} = 0$ otherwise, and (25) holds since $Q^*(t, \mu) \leq \lim_{\mu' \to \infty} Q^*(t, \mu') = -(\tau-t)C + R \leq R$ for all $\mu$ as $Q^*(t, \mu)$ is non-decreasing in $\mu$. Finally, observing

$$\lim_{\mu \to -\infty} (26) = \lim_{\mu \to -\infty} \left( -C + R \left( 1 - \int_{-\infty}^{\mu_t^* - \mu} f(z, 1/(t-1) - 1/t) dz \right) \right)$$

$$= -C \, ,$$

together with bounds (23) and (26), we obtain $\lim_{\mu \to -\infty} Q^*(t-1, \mu) = -C$.

When put together, the base cases and the inductive cases above imply that conditions (i–iii) hold for all $t \in \{1, \ldots, \tau-1\}$—hence $\mu_t^*$ exists for all $t \in \{1, \ldots, \tau-1\}$—which concludes our proof. $\square$

### I.3    PROOF OF PROPOSITION 3

First, we prove the existence of $\mu_t^{\text{greedy}}$ for all $t \in \{0, \ldots, \tau-1\}$ by driving an analytical formula for $V^{(0)}(t, \mu) \doteq V^{\pi^{(0)}}(t, \mu)$. Letting $T_t^{(0)} = T_t^{\pi^{(0)}}$,

$$V^{(0)}(t, \mu) = \mathbb{E}[R \cdot \mathbb{1}\{T_t^{(0)} > \tau\} \cdot \rho(\mu_\tau) - C \cdot (\min\{T_t^{(0)}, \tau\} - t) | \mu_t = \mu]$$

$$= \mathbb{E}[R \cdot \rho(\mu_\tau) - C \cdot (\tau - t) | \mu_t = \mu] \tag{27}$$

$$= -C + \int \mathbb{E}[R \cdot \rho(\mu_\tau) - C \cdot (\tau - t - 1) | \mu_{t+1} = \mu'] d\mathbb{P}(\mu_{t+1} = \mu' | \mu_t = \mu)$$

$$= -C + \int V^{(0)}(t+1, \mu') d\mathbb{P}(\mu_{t+1} = \mu' | \mu_t = \mu)$$

$$= -C + \int V^{(0)}(t+1, \mu') f(\mu' - \mu; 1/t - 1/(t+1)) d\mu' \tag{28}$$

$$= -C + \int V^{(0)}(t+1, \mu+z) f(z; 1/t - 1/(t+1)) dz \, , \tag{29}$$

where (27) holds since $\pi^{(0)}(t, \mu) = \Psi_0$ for all $t$ and $\mu$ hence it is always the case that $T_t^{(0)} = \infty$, and (28) is due to (15).

In the remainder of our proofs, we take $\alpha = 0$ for notational brevity. This is without any loss of generality as, by simply shifting each value function and Q-function by $\alpha/\sqrt{\tau}$ with respect to $\mu$, all of the following arguments would still hold. For $\alpha = 0$, we show that

$$V^{(0)}(t, \mu) = -(\tau - t)C + R \cdot F\left( \frac{\mu}{\sqrt{1/t - 1/\tau}} \right) \tag{30}$$

for all $t \in \{1, \ldots, \tau-1\}$ via mathematical induction. Note that (30) is true for $t = \tau - 1$:

$$V^{(0)}(\tau-1, \mu) = -C + \int V^{(0)}(\tau, \mu+z) f(z; 1/(\tau-1) - 1/\tau) dz$$

$$= -C + R \int \mathbb{1}\{\mu+z > 0\} f(z; 1/(\tau-1) - 1/\tau) dz$$

$$= -C + R \int_{-\mu}^{\infty} f(z; 1/(\tau-1) - 1/\tau) dz$$

$$= -C + R \int_{-\infty}^{\mu} f(z; 1/(\tau-1) - 1/\tau) dz$$

$$= -C + R \int_{-\infty}^{\mu/\sqrt{1/(\tau-1) - 1/\tau}} f(z; 1) dz$$

$$= -C + R \cdot F\left(\frac{\mu}{\sqrt{1/(\tau-1) - 1/\tau}}\right),$$

where $F(x) \doteq F(x; 1)$ is the c.d.f. of the standard Gaussian distribution. Moreover, assuming (30) is true for $t$, it is also true for $t - 1$:

$$V^{(0)}(t - 1, \mu)$$
$$= -C + \int V^{(0)}(t, \mu + z) f(z; 1/t_{-1} - 1/t) dz$$
$$= -(\tau - t + 1)C + R \int F((\mu + z)/\sqrt{1/t - 1/\tau}; 1) f(z; 1/(t-1) - 1/t) dz$$
$$= -(\tau - t + 1)C$$
$$\quad + R \iint_{-\infty}^{(\mu+z)/\sqrt{1/t - 1/\tau}} f(z'; 1) f(z; 1/(t-1) - 1/t) dz' dz$$
$$= -(\tau - t + 1)C$$
$$\quad + R \iint_{-\infty}^{\mu+z} f(z'; 1/t - 1/\tau) f(z; 1/(t-1) - 1/t) dz' dz$$
$$= -(\tau - t + 1)C$$
$$\quad + R \iint \mathbb{1}\{z' \le \mu + z\} f(z'; 1/t - 1/\tau) f(z; 1/(t-1) - 1/t) dz' dz$$
$$= -(\tau - t + 1)C + R \cdot \mathbb{P}_{\substack{Z \sim \mathcal{N}(0, 1/(t-1) - 1/t) \\ Z' \sim \mathcal{N}(0, 1/t - 1/\tau)}} \{Z' \le \mu + Z\}$$
$$= -(\tau - t + 1)C + R \cdot \mathbb{P}_{\frac{Z'-Z}{\sqrt{1/(t-1) - 1/\tau}} \sim \mathcal{N}(0,1)} \left\{ \frac{Z' - Z}{\sqrt{1/(t-1) - 1/\tau}} \le \frac{\mu}{\sqrt{1/(t-1) - 1/\tau}} \right\}$$
$$= -(\tau - t + 1)C + R \cdot F\left(\frac{\mu}{\sqrt{1/(t-1) - 1/\tau}}\right).$$

Therefore, (30) indeed holds for all $t \in \{1, \dots, \tau - 1\}$.

Next, we observe that $V^{(0)}(t, \mu)$ has a root at $\mu = F^{-1}((\tau - t)C/R)\sqrt{1/t - 1/\tau}$ provided that $(\tau - t)C/R \in (0, 1)$, which is the case for all $t \in \{1, \dots, \tau-1\}$ since $\tau C < R$. Moreover, $V^{(0)}(t, \mu)$ is a strictly increasing function in $\mu$. Hence, there exists a unique $\mu_t^{\text{greedy}}$ for all $t \in \{1, \dots, \tau - 1\}$ such that $V^{(0)}(t, \mu_t^{\text{greedy}}) > 0$ and $V^{(0)}(t, \mu) > 0 \iff \mu > \mu_t^{\text{greedy}}$. In other words, $\pi^{\text{greedy}}$ is also a thresholding-type policy as the proposition states.

Finally, we have $V^{(0)}(t, \mu_t^*) = Q^{(0)}(t, \mu_t^*) \le Q^*(t, \mu_t^*) = 0$ hence $\mu_t^* \le \mu_t^{\text{greedy}}$. This is because, by definition, $Q^*(t, \mu) \ge Q^\pi(t, \mu)$ for all $t, \mu$ for any given policy $\pi$, including $\pi^{(0)}$. □

### I.4 PROOF OF PROPOSITION 4

As in the proof of Proposition 3, we take $\alpha = 0$ for notational brevity. Once again, this is without any loss of generality as, by simply shifting each value function and Q-function by $\alpha/\sqrt{\tau}$ with respect to $\mu$, all of the following arguments would still hold. Remember that the formula we derived for $V^{(0)}(t, \mu)$ in (30) holds when $\alpha = 0$.

We start by deriving two bounds on the optimal Q-function $Q^*(t, \mu)$: (i) a lower bound and (ii) an upper bound. For the lower bound, it is sufficient to observe that

$$V^{(0)}(t, \mu) = Q^{(0)}(t, \mu) \le Q^*(t, \mu),$$

which holds since, by definition, $Q^*(t, \mu) \ge Q^\pi(t, \mu)$ for all $t, \mu$ for any given policy $\pi$.

For the upper bound, we use mathematical induction to show that $Q^*(t, \mu) \le (\tau - t - 1)C + V^{(0)}(t, \mu)$. First, for the base case of $\tau - 1$,

$$Q^*(\tau - 1, \mu) = -C + \int V^*(\tau, \mu + z) f(z; 1/t - 1/t+1) dz \qquad (31)$$
$$= -C + \int \mathbb{1}\{\mu + z > \alpha\} f(z; 1/t - 1/t+1) dz$$

$$= -C + \int V^{(0)}(\tau, \mu + z) f(z; 1/t - 1/t+1) dz$$
$$= V^{(0)}(\tau - 1, \mu) \,, \tag{32}$$

where (31) is due to (16), and (32) is due to (29). Then, for the inductive case, assuming $Q^*(t, \mu) \leq (\tau - t - 1)C + V^{(0)}(t, \mu)$,

$$Q^*(t-1, \mu) = -C + \int \max\{0, Q^*(t, \mu + z)\} f(z, 1/(t-1) - 1/t) dz \tag{33}$$
$$\leq \int Q^*(t, \mu + z) f(z, 1/(t-1) - 1/t) dz \tag{34}$$
$$\leq (\tau - t - 1)C + \int V^{(0)}(t, \mu + z) f(z, 1/(t-1) - 1/t) dz$$
$$= (\tau - t)C + V^{(0)}(t-1, \mu + z) \,,$$

where (33) is due to (17), and (34) holds since $-C \leq Q^*(t, \mu)$ implies that $\max\{0, Q^*(t, \mu)\} \leq \max\{C + Q^*(t, \mu), Q^*(t, \mu)\} \leq C + Q^*(t, \mu)$.

Define $\mu_t^+$ and $\mu_t^-$ such that

$$V^{(0)}(t, \mu_t^+) = 0 \iff \mu_t^+ = F^{-1}\left((\tau - t)\frac{C}{R}\right)\sqrt{\frac{1}{t} - \frac{1}{\tau}}$$

$$(\tau - t - 1)C + V^{(0)}(t, \mu_t^-) = 0 \iff \mu_t^- = F^{-1}\left(\frac{C}{R}\right)\sqrt{\frac{1}{t} - \frac{1}{\tau}} \,,$$

which we are able to write in closed form using the formula we derived for $V^{(0)}(t, \mu)$ in (30) during the proof of Proposition 3.

By definition, $\mu_t^{\text{greedy}} = \mu_t^+$. Moreover, (i) $V^{(0)}(t, \mu_t^*) \leq Q^*(t, \mu_t^*) = 0 = V^{(0)}(t, \mu_t^+)$ due to our lower bound, hence $\mu_t^* \leq \mu_t^+$ (remember that $V^{(0)}(t, \mu)$ was a strictly increasing function in $\mu$), and (ii) $(\tau - t - 1)C + V^{(0)}(t, \mu_t^-) = 0 = Q^*(t, \mu_t^*) \leq (\tau - t - 1)C + V^{(0)}(t, \mu_t^*)$ due to our upper bound, hence $V^{(0)}(t, \mu_t^-) \leq V^{(0)}(t, \mu_t^*)$ meaning $\mu_t^- \leq \mu_t^*$. Putting together these facts, and also the fact that $\mu_t^* \leq \mu_t^{\text{greedy}}$, we obtain $|\mu_t^* - \mu_t^{\text{greedy}}| \leq \mu_t^+ - \mu_t^-$ as the proposition states. $\qquad\square$

## J  BENCHMARKING ALGORITHMS

---
**Algorithm 2** Adaptive Enrichment, Futility Stopping with Bayes-OCP, Greedy Bayes-OCP
---
1: Initialize $\mu_x$ and $\sigma_x^2$ for all $x \in \mathcal{X}$
2: $X \leftarrow \mathcal{X}, \quad t \leftarrow 0, \quad \mathcal{D}_0 \leftarrow \varnothing$
3: Start experiment $\psi = (\mathcal{X}, \tau, \rho)$
4: **loop:**
5: $\quad t \leftarrow t + 1$
6: $\quad$ Observe $x_t, y_t$
7: $\quad \mathcal{D}_t \leftarrow \mathcal{D}_{t-1} \cup \{x_t, y_t\}$
8: $\quad 1/\sigma_{x_t}^2 \leftarrow 1/\sigma_{x_t}^2 + 1$
9: $\quad \mu_{x_t} \leftarrow \mu_{x_t} + (y_t - \mu_{x_t})\sigma_{x_t}^2$
10: $\quad X' \leftarrow \varnothing$
11: $\quad$ **while** $X \setminus X' \supset \varnothing$:
12: $\quad\quad x^* \leftarrow \text{argmax}_{x \in X \setminus X'} \mathbb{E}_{\theta_x \sim \mathcal{N}(\mu_x, \sigma_x^2)}[\mathcal{G}^{(0)}(X' \cup \{x\}; \{\theta_x\})]$
13: $\quad\quad$ **if** $\mathbb{E}_{\theta_x \sim \mathcal{N}(\mu_x, \sigma_x^2)}[\mathcal{G}^{(0)}(X' \cup \{x^*\}; \{\theta_x\})] > \mathbb{E}_{\theta_x \sim \mathcal{N}(\mu_x, \sigma_x^2)}[\mathcal{G}^{(0)}(X'; \{\theta_x\})]$:
14: $\quad\quad\quad X' \leftarrow X' \cup \{x^*\}$
15: $\quad\quad$ **else:**
16: $\quad\quad\quad$ **break**
17: $\quad$ **if** *Adaptive Enrichment*
$\quad\quad$ **and** $t = \tau/2$ **and** $\mathbb{E}_{\theta_x \sim \mathcal{N}(\mu_x, \sigma_x^2)}[\mathcal{G}^{(0)}(X'; \{\theta_x\})] > \mathbb{E}_{\theta_x \sim \mathcal{N}(\mu_x, \sigma_x^2)}[\mathcal{G}(X, \mathcal{D}_t; \{\theta_x\})]$:
18: $\quad\quad X \leftarrow X', \quad t \leftarrow 0, \quad \mathcal{D}_0 \leftarrow \varnothing$
19: $\quad\quad$ Start a new experiment $\psi = (X, \tau, \rho)$
20: $\quad$ **if** *Greedy Bayes-OCP* **and** $\mathbb{E}_{\theta_x \sim \mathcal{N}(\mu_x, \sigma_x^2)}[\mathcal{G}^{(0)}(X'; \{\theta_x\})] > \mathbb{E}_{\theta_x \sim \mathcal{N}(\mu_x, \sigma_x^2)}[\mathcal{G}(X, \mathcal{D}_t; \{\theta_x\})]$:
21: $\quad\quad X \leftarrow X', \quad t \leftarrow 0, \quad \mathcal{D}_0 \leftarrow \varnothing$
22: $\quad\quad$ Start a new experiment $\psi = (X, \tau, \rho)$
23: $\quad$ **if** *Futility Stopping with Bayes-OCP* **and** $\mathbb{P}_{\theta_x \sim \mathcal{N}(\mu_x, \sigma_x^2)}\{\mathcal{G}^{(0)}(\varnothing; \{\theta_x\}) > \mathcal{G}(X, \mathcal{D}_0; \{\theta_x\})\} > \beta$:
24: $\quad\quad$ Stop all experimentation
---

## K    GLOSSARY OF TERMS AND NOTATION

| Term | Notation | Description |
|---|---|---|
| Experiment | – | Conducted to confirm efficacy of an intervention, e.g. a new treatment in clinical trials, or a new recommendation policy in online advertisement |
| Subject | – | Individual participant of an experiment, e.g. patients in a clinical trial, or customers in online advertisement |
| Population | $X \subseteq \mathcal{X}$ | Collection of subjects that all share the same qualities, e.g. all female patients in a clinical trial, or all customers with the same preferences in online advertisement |
| Atomic-population | $x \in \mathcal{X}$ | Indivisible populations |
| Propensities | $\eta_x$ | The probability that a subject being from atomic-population $x$ |
| | $\eta_X$ | The probability that a subject being from population $X$ |
| | $\eta_{x\|X}$ | The probability that a subject being from atomic-population $x$ conditioned on the fact that they from population $X$ |
| Outcome distribution | $\Omega_x$ | Distribution of outcomes that is indicative of the effect of the intervention of interest for atomic-population $x$ |
| Mean outcomes | $\theta_x$ | Expected outcome, i.e. the effect of the intervention of interest, for atomic-population $x$ |
| | $\bar{\theta}_X$ | Expected outcome for population $X$ |
| Experiment design | $\psi = (X, \tau, \rho)$ | Target population $X$, sample horizon $\tau$, and success criterion $\rho$ that characterize an experiment |
| Viable experiment designs | $\Psi$ | Experiment designs that can potentially be followed by a meta-experimenter |
| Meta-experimenter | – | The decision-making agent that decides when to run experiments according to which experiment design in $\Psi$ |
| Sample/time horizon | $\tau$ | An experiment is terminated when $t = \tau$ |
| Success criterion | $\rho$ | An experiment is declared a success if $\rho(\mathcal{D}_\tau) = 1$ |
| Online dataset | $\mathcal{D}_t$ | Data collected by an ongoing experiment at time step $t$ |
| | $\mathcal{D}_t^i$ | Data collected by the $i$-th experiment run by the meta-experimenter at time step $t$ |
| Aggregate dataset | $\bar{\mathcal{D}}_t^i$ | Collective data collected by all experiments up to time step $t$ of the $i$-th experiment |
| – | $T^i$ | Number of time steps for which the $i$-th experiment is conducted until it was stopped or its time horizon was reached |
| Cost | $C_\psi$ | Cost incurred per time step by running experiment $\psi$ |
| Reward | $R_\psi$ | Reward received if experiment $\psi$ is successful |
| Utility | $G$ | Sum of costs and rewards received after all experimentation is concluded |
| Policy | $\pi$ | Decision-making policy of the meta-experiment |
| Optimal policy | $\pi^*$ | The optimal policy that maximizes utility $G$ in expectation |
| Greedy policy | $\pi^{\text{greedy}}$ | See Section 3 |
| Test statistic | $\mu_t$ | In the simplified case in Section 3, the empirical mean outcome |
| Value function | $V^\pi(t, \mu)$ | The expected utility of following policy $\pi$ when $\mu_t = \mu$ |
| Q-function | $Q^\pi(t, \mu)$ | The expected utility of following policy $\pi$ after conducting the ongoing experiment for one more time step when $\mu_t = \mu$ |
| – | $T_t^\pi$ | The first time step at or after time step $t$ that policy $\pi$ decides to stop all experimentation |
| Optimal value function | $V^*$ | The value function associated with $\pi^*$ |
| Optimal Q-function | $Q^*$ | The Q-function associated with $\pi^*$ |
| Thresholds | $\mu_t^*$ | Decision-making thresholds associated with $\pi^*$ |
| | $\mu_t^{\text{greedy}}$ | Decision-making thresholds associated with $\pi^{\text{greedy}}$ |

| Term | Notation | Description |
|------|----------|-------------|
| Conditional power function | $\mathcal{P}(X, \mathcal{D}_t; \{\theta_x\})$ | The probability of a hypothesis test being successful conditioned on mean outcomes $\{\theta_x\}$ |
| Expected utility function | $\mathcal{G}(X, \mathcal{D}_t; \{\theta_x\})$ | The expected utility of fully committing to an experiment and waiting until it terminates when the experiment targets population $X$, is currently at time step $t$, and collected dataset $\mathcal{D}_t$ |
| Posteriors | $\mathcal{N}(\mu_x, \sigma_x^2)$ | Posterior distributions over mean outcomes $\{\theta_x\}$ maintained by Bayes-OCP such that $\theta_x \vert \bar{\mathcal{D}} \sim \mathcal{N}(\mu_x, \sigma_x^2)$ |

