# OpenReview forum: "When to Make and Break Commitments?"
_ICLR.cc/2023/Conference — ICLR 2023 poster_

### Official Review · Reviewer_tD32 · 2022-10-25

**Confidence:** 3
**Correctness:** 4
**Technical Novelty And Significance:** 2
**Empirical Novelty And Significance:** 2
**Recommendation:** 6

**Clarity, Quality, Novelty And Reproducibility:**

*Clarity & Quality:*
The paper is clearly written and easy to follow. The set of results represent a good starting point to exploring this problem. At the same time, the experimental results could be expanded, while the analysis could focus on more general, less constrained settings. E.g., are there other instances of the problem setting for which one can obtain similar results? If so, those results could be informative.
---
*Novelty & Reproducibility:*
The problem setting appears to be novel and, in my opinion, this is the main strength of the paper. The results look reproducible: the paper documents well the experimental testbed, while the appendix includes proofs of the claims from the main text.

**Strength And Weaknesses:**

**Strengths**:
- Arguably, the main strength of this paper is the novelty of the paper is the problem setting. To my knowledge, this particular setting has not been studies in the literature.
- The characterization results on a simpler instance of the problem setting are intuitive, but provide insights that one can use to design effective algorithms. These results also inspired the design of the propose algorithm for optimally deciding on commitments.
---
**Weaknesses**:
- The more complex, and presumably more interesting setting is not formally analyzed. This makes the set of characterization results somewhat incomplete.
- The algorithm assumes that the outcomes of interest come from a specific distribution, which does not affect the soundness of the algorithm, but may affect its performance.
- The proposed algorithm doesn't seem to have any provable guarantee that it converges to an optimal solution, and it is not clear how it would fare against e.g. policies learned via RL with function approximation. The paper only reported results with relatively simple RL baseline.
- Benchmarks appear to be rather simple with only two atomic population and carefully chosen parameters. Although the experimental results indicate that the proposed approach is more effective than the baselines, more extensive experimentation would be quite useful. For example, the experimental setup where the distributional assumption on the outcome could fail to hold would be informative about the robustness of the proposed approach.

**Summary Of The Paper:**

This paper studies the setting of optimally breaking commitments. In particular, it considers a decision making problem in which a decision maker has to decide on whether to break a commitment and if so, whether to select an alternative commitment or stop the making commitments. The paper formally characterizes a simplified version of the problem, which leads to three key insights that are later on used to design a novel algorithm for a more general setting. This novel algorithm is validated in a simulation-based test-bed.

**Summary Of The Review:**

While I believe that this paper studies an interesting topic, and provides an interesting set of initial results, the current results look preliminary at the first glance and could be further extended. More concretely, the theoretical analysis focuses on a simpler instance of the problems setting, while the experiments are based on simple benchmarks. In my opinion, the paper could greatly benefit from adding experiments that tests if the proposed algorithm is brittle to the deviation from the distributional assumption it relies on, and potentially adding more sophisticated baselines.

---

> ### Author Response · Authors · 2022-11-10
> **Response to Reviewer tD32 [Part 4/4]**
>
> **Convergence to an Optimal Solution**
>
> We would like to clarify that, by no means we claim that Bayes-OCP is an optimal solution to the optimal commitment problem, or that it converges to the optimal solution (Bayes-OCP does not “converge” to any solution as it does not involve a training phase like most RL algorithms do). Bayes-OCP is a heuristic solution that is only expected to perform well on the basis that it shares some key characteristics with the optimal solution—namely Bayes-OCP is optimistic and it is also increasingly less optimistic.
>
> We agree with the reviewer that perhaps showing an upper bound on the suboptimality gap of Bayes-OCP would have been informative. However, we believe this to be beyond the current scope of our paper, which not only formulates a brand new problem, but also provides a preliminary theoretical analysis of that problem—in particular, showing the inadequacy of standard RL-based techniques in solving it—and proposes an initial solution.

---

> > ### Author Response · Authors · 2022-11-14
> > **Dear Reviewer tD32**
> >
> > Once again, thank you for your invaluable feedback. We were wondering whether our response and the revised manuscript addressed your concerns. If you have any additional comments, please let us know, we would be happy to address them.

---

> > > ### Comment · Reviewer_tD32 · 2022-11-26
> > > **Thank you for your response**
> > >
> > > The authors' response addresses some of my concerns, so I have updated my score accordingly.

---

> ### Author Response · Authors · 2022-11-10
> **Response to Reviewer tD32 [Part 3/4]**
>
> **Limitations of the Formal Analysis**
>
> We consider a simplified instance during our formal analysis of the problem in Section 3 mainly to provide a better intuition of the optimal commitment problem before we start tackling the more general and the more complex case. Although the results given as Proposition 2-4 are technically limited to this simplified instance, they still provide a general sense of what it means for an algorithm to be optimistic (or for an algorithm to be increasingly less optimistic) in the context of optimal commitment in a mathematically precise manner. We make use of this intuition in designing Bayes-OCP in Section 4 and reference the relevant propositions when explaining our design. We also validate the intuition we gain regarding optimism with experiments in Section 5 (see Figure 4, where we reference back to Proposition 3).
>
> It should also be emphasized that the implication of Proposition 1 is not limited to the simplified instance considered in Section 3. Proposition 1 gives a counter-example to demonstrate the existence of at least one problem instance where value and Q-functions are non-convex. Regardless of whether that instance also happens to be one of the simplified instances considered in Section 3, the point that optimal commitment is a non-convex problem and convex function approximation is not a suitable approach to solving the optimal commitment problem still stands even for the most general setting. In Appendix A, we discuss optimal commitment in detail from a reinforcement learning perspective. There, we formulate value and Q-functions more generally and re-iterate the significance of Proposition 1 in terms of finding an RL-based solution to optimal commitment.

---

> ### Author Response · Authors · 2022-11-10
> **Response to Reviewer tD32 [Part 2/4]**
>
> **Supplementary Experiments**
>
> First, we would like to highlight that we already provide a variety of supplementary results that explore different settings, environments, and benchmarks (see the paragraph title “supplementary results” in Section 5, also see Appendices A.1, B.1, C, and D).
>
> We have also now included even more supplementary results as the reviewer has kindly suggested. In a **new Appendix E**, we consider environments with more than two atomic-subpopulations (specifically, $|\mathcal{X}|=10$). The results are given in the table below (see Table 7 in the revised paper), we still see that Bayes-OCP performs the best.
>
> | Algorithms: | RCT    | Adaptive Enrichment | Futility Stopping | Greedy Bayes-OCP | **Bayes-OCP** |
> |-------------|:------:|:-------------------:|:------------------------------:|:----------------:|:-------------:|
> | Utility     | 8.0    | 143.9               | 141.0                          | 40.3             | **172.4**     |
> | FWER        | 0.0\%  | 0.0\%               | 0.0\%                          | 0.0\%            | 0.0\%         |
> | Switches    | 0.0    | 0.4                 | 0.5                            | 1.0              | 0.6           |
> | Success     | 60.8\% | 71.0\%              | 51.2\%                         | 15.6\%           | 63.2\%        |
> | T-to-S      | 600.0  | 678.9               | 600.0                          | 648.5            | 647.5         |
> | T-to-F      | 600.0  | 672.7               | 130.4                          | 9.3              | 200.5         |
>
> In the **revised Appendix A.1**, we compare Bayes-OCP against a more sophisticated RL-based benchmark, specifically we follow the approach by a recent paper (Ni et al. "Recurrent model-free RL can be a strong baseline for many POMDPs," ICML, 2022) for solving complex classes of POMDs, which the optimal commitment problem is one of as we already highlight in Appendix A. The results are given in the table below (see Table 4 in the revised paper), we see that Bayes-OCP performs better.
>
> | Algorithms: | Futility Stopping w/ Ni et al. 2022 | Futility Stopping w/ Bayes-OCP |
> |-------------|:------------------------------------:|:------------------------------:|
> | Utility     | 78.8                                 | **150.0**                      |
> | FWER        | 0.0%                                 | 0.1%                           |
> | Switches    | 0.7                                  | 0.5                            |
> | Success     | 24.2%                                | 45.4%                          |
> | T-to-S      | 600.0                                | 600.0                          |
> | T-to-F      | 23.6                                 | 57.6                           |

---

> ### Author Response · Authors · 2022-11-10
> **Response to Reviewer tD32 [Part 1/4]**
>
> Thank you for your thoughtful comments and suggestions! Please find our answers as follows, along with corresponding updates to the revised submission:
>
> **Experiments with Misspecified Distributions**
>
> We agree with the reviewer that Bayes-OCP needs evaluation when its distributional assumption is invalidated. *That is why we have already presented such results in Appendix C.* Briefly summarizing the discussion there, we describe a new environment where outcomes are binary ($y\in\{-1,1\}$) instead of continuous, and have a categorical distribution rather than a Gaussian distribution. Despite this, we leave the implementation of Bayes-OCP unchanged, which still assumes outcomes to be normally distributed. We repeat our main experiments in Section 5 in this newly described environment and still see that Bayes-OCP performs the best among all our benchmarks (see Table 6).

---

### Official Review · Reviewer_mtcX · 2022-10-27

**Confidence:** 3
**Correctness:** 4
**Technical Novelty And Significance:** 4
**Empirical Novelty And Significance:** 2
**Recommendation:** 6

**Clarity, Quality, Novelty And Reproducibility:**

Paper is clearly written, results seem high quality and the problem being considered is new as far as I know.

**Strength And Weaknesses:**

Strengths

— Well motivated: The paper describes application of this problem formulation to multiple domains and motivates the applicability of these techniques well.

– Well written: The paper is well written overall and is easy to follow. Inclusion of warm-up section for example is useful in building understanding.

– New formulation: The OCP problem formulation seems novel to the best of my knowledge.

– Good results: The proposed methodology and algorithm is shown to perform well.

Weaknesses

– Theoretical guarantees on algorithm performance would be valuable

– Related work: There is some related work on queuing theory and optimization in cloud computing (completely different application domain) such as “Efficient Straggler Replication in Large-scale Parallel Computing” by Wang et al. which focuses on the problem of when to kill existing tasks and relaunch. It may be interesting to discuss if these techniques are relevant or if there are fundamental shortcomings that prohibit their use for the OCP problem.


**Summary Of The Paper:**

This paper considers a problem setup in which decision makers launch experiments (say over target population of agents) and have to wait a period of time before receiving the outcomes. These outcomes may or may not be conclusive, and accordingly the experiment launched may or may not be useful. If found inconclusive, it may be useful to relaunch the experiment with a new sample population. This paper studies the problem of determining when it is optimal to continue or break the commitment to a particular instance of an experiment.

The paper formulates this as an “Optimal COmmitment Problem (OCP)” and provide some theoretical analysis. The paper also proposes a practical algorithm for solving the OCP problem and evaluates it empirically in the context of clinical trials.


**Summary Of The Review:**

New problem domain, well-motivated, first results in terms of theory and algorithm.

Good paper overall

---

> ### Author Response · Authors · 2022-11-10
> **Response to Reviewer mtcX [Part 2/2]**
>
> **Related Work on Queuing Theory and Optimization in Cloud Computing**
>
> We thank the reviewer for bringing this strand of literature to our attention. Deciding when to kill an existing task and relaunch it (Wang et al.) is certainly related to optimal stopping/switching. However there, the focus is on reasoning about when a stochastic event (i.e. successful completion of a computational task) will occur *without any extra information* other than the fact that the event of interest has not occurred yet.
> In contrast, in our setting, the decision maker needs to process a *streaming set of samples* to reason about the random outcome of an event that is scheduled to happen at a deterministic time point (here, the event is an experiment reaching its conclusion).
> This means that our problem and Wang et al. have a completely different information structure.
>
> More formally, we observe samples $y_t$ that are informative of whether $\rho(\mathcal{D}_\tau)=1$ when $\tau$ is a fixed variable. In contrast, the problem in Wang et al. would correspond to the setting where $\tau$ is a random variable with a known distribution and $\rho=1$ always holds (hence no need to observe any samples $y_t$).
>
> Among optimal stopping/switching problems, the structure of our problem is more closely related to sequential hypothesis testing, hence why, our related work primarily focuses on SHT. However, we now highlight Wang et al. (among others) as a related problem in a **new extended related work in Appendix H**.

---

> > ### Author Response · Authors · 2022-11-14
> > **Dear Reviewer mtcX**
> >
> > Once again, thank you for your invaluable feedback. We were wondering whether our response and the revised manuscript addressed your concerns. If you have any additional comments, please let us know, we would be happy to address them.

---

> > ### Comment · Reviewer_mtcX · 2022-11-17
> > **Thank you**
> >
> > Thank you for responding to both the comments/ questions listed under "weaknesses". I read through the responses and am satisfied with the authors' response.

---

> > > ### Author Response · Authors · 2022-11-18
> > > **Re: Thank you**
> > >
> > > We are happy that you were satisfied with our response. If we have managed to address all your concerns, we would like to kindly ask you to consider adjusting your score, taking the rebuttal and the revised manuscript into account.

---

> ### Author Response · Authors · 2022-11-10
> **Response to Reviewer mtcX [Part 1/2]**
>
> Thank you for your thoughtful comments and suggestions! Please find our answers as follows, along with corresponding updates to the revised submission:
>
> **Theoretical Guarantees on Performance**
>
> While our theoretical results motivate the general use of an optimistic decision rule, they indeed do not provide any guarantees about the performance of the specific rule we propose as part of Bayes-OCP. We agree with the reviewer that perhaps showing an upper bound on the suboptimality gap of Bayes-OCP would have been informative. However, we believe this to be beyond the current scope of our paper, which not only formulates a brand new problem, but also provides a preliminary theoretical analysis of that problem and proposes an initial solution using the insight gained from that analysis. As such, we now mention in a **new Appendix E** the analysis of our algorithm’s performance as a future research direction.

---

### Official Review · Reviewer_jpJi · 2022-11-03

**Confidence:** 2
**Correctness:** 4
**Technical Novelty And Significance:** 3
**Empirical Novelty And Significance:** 3
**Recommendation:** 8

**Clarity, Quality, Novelty And Reproducibility:**

**Clarity**

Overall, I believe that the paper is of high quality, very well written an well organized.

**Quality**

Although I haven’t checked the proofs in the appendices in detail, the results in the paper strike me as reasonable and the presentation as technically sound, in general.

**Novelty**

As mentioned earlier, the paper is somewhat outside my area of research, so I am not properly able to assess the originality of the work. However, from the discussion in the paper, the contributions appear novel and impactful.

**Reproducibility**

The paper provides a discussion on the reproducibility of the results in the paper. I believe that, in general, the results are reproducible.

**Details Of Ethics Concerns:**

In my view, the paper properly discusses ethical implications of the proposed research.

**Strength And Weaknesses:**

The paper is somewhat outside my comfort area, so it is possible that I misjudge some of the contributions of the paper.

**Strengths:**

The paper is very well written, and a pleasure to read. It makes an excellent work of motivating and setting up the problem, the contributions are clearly exposed and presented in a constructive and intuitive manner. The experiments, although involving mostly simulation, do support the claims of the paper adequately.

**Weaknesses:**

I have, perhaps, two main complaints regarding the paper. The first is that I would have liked to grasp more clearly the relation between the proposed work and other decision-theoretic problems. The paper properly provides the connection between the proposed framework and POMDPs, and also establishes early in the discussion that the OCP problem is, at its essence, one of exploration vs exploitation. This said, it would seem to me that approaches based on sequential prediction/multi-armed bandits (particularly using experts) could also be relevant alternatives to consider, but I found little discussion on this problem. Since $\mathcal{X}$ is finite, an approach that seeks to identify which $x\in\mathcal{X}$ to select that maximizes the utility would be a relevant competitor. I’m probably missing something here, but some discussion on this would certainly help to clarify my possible confusion.

My second complaint is more on form than on content. Personally, I found that laying out the tables and figures in the middle of the text to be disruptive while reading the paper. I realize that this is probably done for a matter of space, but perhaps moving some of the tables/plots to the appendix may help (for example, Figs. 3 and 4 can probably make it to the appendix with no significant loss in the exposition).

**Summary Of The Paper:**

The paper introduces the _optimal commitment problem_, where an agent must decide, at each time step $t$, whether to continue running the current experiment (i.e., _commit_ to the current experiment), or terminate the current experiment, switching to a more promising experiment (or to no experiment at all, if one does not exist). The paper first analyzes the simple case, where the decision is just whether to continue or stop the current experiment. The paper establishes that the optimal policy, in this case, is a threshold policy (i.e., if the outcome of the current experiment is above some threshold continue, otherwise stop) that becomes increasingly close to a greedy policy as the experiment approaches its end.

The paper then proposes Bayes-OCP, an algorithm that maintains a posterior distribution over the mean outcome for each possible atomic population $x\in\mathcal{X}$. Then, at each iteration, assesses whether switching from the current population $X\subset\mathcal{X}$ to an alternative population $X’\subset\mathcal{X}$ (including the “empty” population) is likely to lead to a higher utility. In that case, the agent stops the current experiment and starts a new experiment involving the population $X’$.

The results in the paper show that, when compared with alternative approaches, Bayes-OCP strikes a good balance between optimism (i.e., not stopping the experiments too soon) and timeliness (not stopping the experiments too late).

**Summary Of The Review:**

The paper is, in my opinion, very well written and provides a novel and interesting contribution. I would like to understand better the applicability of multi-armed bandit problems to the setting considered in the paper.

---

> ### Author Response · Authors · 2022-11-10
> **Response to Reviewer jpJi**
>
> Thank you for your thoughtful comments and suggestions! Please find our answer as follows, along with corresponding updates to the revised submission:
>
> **Relationship to Multi-armed Bandits**
>
> The optimal commitment problem is indeed similar to a multi-armed bandit problem in some aspects: Like arms in a MAB problem, each experiment design $\psi$ has a random utility given by $R\_{\psi}\cdot \rho(\mathcal{D}\_{\tau})-\tau C\_{\psi}$, where $\mathcal{D}\_{\tau}$ is the source of randomness, and the distribution of this utility is unknown. Also, like in a MAB problem, the overall goal is to sequentially select experiment designs (cf. arms) that yield the maximum cumulative utility. The main difference between the two problems is that, in a MAB problem, selecting an arm immediately reveals a sample from its random utility, while in optimal commitment, running an experiment $\psi$ just for one time step only incurs a cost of $C_{\psi}$; observing a full sample of its random utility requires the experiment to be run until its completion for $\tau$ consecutive time steps, without selecting any other experiment design in the meantime.
>
> One can naively apply a MAB algorithm by viewing each viable experiment design as  a unique arm, and by running experiments/arms selected by the algorithm until their completion to observe full samples from their unknown utility distributions. However, this obviously side steps the main question we want to answer in optimal commitment: When can we abandon a commitment---in this case, the decision to run an experiment/arm selection until its completion---before fully observing its outcome? Looking at optimal commitment from a MAB perspective reveals that there are two explore-exploit dilemmas present in optimal commitment: One is with respect to which experiment to select next, and the other is with respect to when to preemptively stop the current experiment (i.e. breaking a commitment). MAB algorithms address the former dilemma but not the latter.
>
> We have now included this discussion regarding the relationship of optimal commitment to multi-armed bandit problems in a **new Appendix H**.

---

> > ### Comment · Reviewer_jpJi · 2022-11-17
> > **Response to the author's rebuttal**
> >
> > I thank the authors for the clarification. I am happy with the response.

---

### Official Review · Reviewer_1tQ1 · 2022-11-05

**Confidence:** 3
**Correctness:** 4
**Technical Novelty And Significance:** 3
**Empirical Novelty And Significance:** 3
**Recommendation:** 8

**Clarity, Quality, Novelty And Reproducibility:**

In general, I find the paper relatively easy to follow. The results are also solid and cover many different facets ranging from theory to practice. The framework and algorithm are also new to my knowledge, and the experiment setup is also relatively clear for reproduction. These being said, there are several weaknesses as mentioned above. In addition, I have some questions and suggestions for the authors, as listed below.
* Is it true that in Bayes-OCP, we assume that we know $R_X$ and $C_X$ for every subpopulation $X$, but we don't know $\theta_X$? It would be helpful to more clearly state what are known and what are not.
* In line 15 of Bayes-OCP, should there be a reset of $t$ to $0$, or should $\mathcal{D}_0$ be $\mathcal{D}_t$ instead? Otherwise when going back to line 5 for the next loop, the assignment of $\mathcal{D}_0$ is meaningless.
* When comparing different algorithms, samples and runtime are not mentioned. These should also be discussed for fairer comparisons.
* In Table 3, it would be helpful to highlight other metrics (in addition to utility) as well, especially given that some metrics are the lower the better while the others are the higher the better if I understand correctly.



**Details Of Ethics Concerns:**

The authors have addressed the ethical concerns well enough in their ethics statement.

**Strength And Weaknesses:**

Strengths:
* The theoretical analysis of the simplified OCP with a single commitment is insightful, which not only motivates the general algorithm but also provides a better understanding of the OCP framework.
* The numerical experiments are relatively thorough and the discussion about green, amber and red instances are also inspiring.

Weaknesses:
* The proposed algorithm, Bayes-OCP can only handle experiment designs with fixed time horizon $\tau$ and fixed success criterion $\rho$. The only decision variable becomes the targeting subpopulation $X\in\mathcal{X}$. This limits the applicability and practicality.
* The numerical experiments are conducted only on a synthetic/simulation environment with only two candidate targeting atomic populations. It would be more convincing if some experiments on more candidate targeting atomic populations and real-world datasets are included.



**Summary Of The Paper:**

This paper studies the problem of online decision making, in which the decision maker decides whether to commit to long-term actions until success or failure, or stop it and/or switch to an alternative. The authors formulate it as an optimal commitment problem (OCP), which is a new type of optimal stopping/switching problem. For a simplified case of OCP with a single commitment, the authors provide theoretical analysis to obtain the characteristics of the optimal solution, and then propose a practical algorithm called Bayes-OCP to solve the general problems. Numerical experiments are also provided to validate the superiority of the proposed framework compared to related existing methods in the literature.

**Summary Of The Review:**

Overall, I find the paper is generally well-written and contains some good results both in theory and practice. But there are also several points for improvements as detailed above.

---

> ### Author Response · Authors · 2022-11-10
> **Response to Reviewer 1tQ1 [Part 3/3]**
>
> **Answers to Questions Regarding Clarity**
>
> * $R_X$ and $C_X$ (more generally $R_{\psi}$ and $C_{\psi}$) are known, which we have now clarified in our problem formulation, and $\theta_x$ is unknown for each atomic-population $x$ as already mentioned under “Objective” in Section 2.
> * Indeed, the time index $t$ should be reset in line 15 of Algorithm 1. We have now updated the algorithm accordingly.
> * We do report the average number of samples until a successful outcome (Time-to-Success, T-to-S) or an unsuccessful outcome (Time-to-Failure, T-to-F) for each benchmark in Table 3. We also mention that $\tau=600$, which is the number of samples needed for a successful hypothesis test, in the preamble of Section 5.
> * In a **new Appendix G**, we now discuss other metrics reported in Table 3 (not just utility and FWER) like the reviewer has kindly suggested.

---

> > ### Comment · Reviewer_1tQ1 · 2022-11-27
> > **Response to the authors' responses**
> >
> > I would like to sincerely thank the authors for the detailed explanations in their responses, and I'm very glad to see the improved draft and additional experiments. But given that there are not intermediate scores, unfortunately I cannot further increase my score (which would go to the full score). Nevertheless, for the reference of the (senior) area chairs and program chairs, I would like to confirm again that the overall quality of the current draft has been obviously further improved.

---

> ### Author Response · Authors · 2022-11-10
> **Response to Reviewer 1tQ1 [Part 2/3]**
>
> **Additional Numerical Experiments**
>
> As we have discussed under “Environments” in Section 6, evaluating the performance of a meta-experiment requires us to consider a variety of environments with different ground-truth effects (like green vs. amber vs. red environment instances). This requires us to use simulations with treatment effects that can freely be adjusted rather than real-world data, which would only capture one true treatment effect.
>
> However, like the reviewer has kindly suggested, simulations with larger numbers of atomic-populations are possible and would indeed be insightful. We have now included **new experimental results in Appendix F** that feature environments with more than two atomic populations (specifically, $|\mathcal{X}|=10$). Results are given in the table below (see Table 7 in the revised paper), we see that Bayes-OCP still performs the best. These results confirm that a greedy approximation is suitable in identifying candidate experiments when the number of atomic-populations is large.
>
> | Algorithms: | RCT    | Adaptive Enrichment | Futility Stopping | Greedy Bayes-OCP | **Bayes-OCP** |
> |-------------|:------:|:-------------------:|:------------------------------:|:----------------:|:-------------:|
> | Utility     | 8.0    | 143.9               | 141.0                          | 40.3             | **172.4**     |
> | FWER        | 0.0\%  | 0.0\%               | 0.0\%                          | 0.0\%            | 0.0\%         |
> | Switches    | 0.0    | 0.4                 | 0.5                            | 1.0              | 0.6           |
> | Success     | 60.8\% | 71.0\%              | 51.2\%                         | 15.6\%           | 63.2\%        |
> | T-to-S      | 600.0  | 678.9               | 600.0                          | 648.5            | 647.5         |
> | T-to-F      | 600.0  | 672.7               | 130.4                          | 9.3              | 200.5         |

---

> ### Author Response · Authors · 2022-11-10
> **Response to Reviewer 1tQ1 [Part 1/3]**
>
> Thank you for your thoughtful comments and suggestions! Please find our answers as follows, along with corresponding updates to the revised submission:
>
> **Limitations of the Proposed Algorithm**
>
> As the reviewer has rightly pointed out, when proposing a solution to the optimal commitment problem, we have chosen only to focus on the selection of a target population $X\subseteq\mathcal{X}$. The reason behind this choice is that we believe the target population of an experiment to be the most critical design dimension to adjust adaptively. As we have already highlighted in our introduction, experiments with inflexible target populations can be problematic when responses to the treatment of interest are highly heterogeneous.
>
> Although our focus has been the target population $X$, the high-level strategy of our proposed algorithm should still be applicable to adapting other design dimensions, namely $\tau$ and $\rho$. At a high-level, Bayes-OCP first identifies a candidate experiment and then compares the identified experiment to the ongoing experiment in an optimistic manner. Regardless of the given set of viable experiment designs $\Psi$, one could still follow the same strategy; the only complication would be to adapt how candidate experiments are identified depending on what design dimension varies across experiment designs provided in $\Psi$.
>
> For instance, when experiment designs vary in terms of $X$ (as we focused on), a combinatorial search is required to identify good candidate experiments, for which we have proposed a greedy strategy. When experiment designs vary in terms of $\rho$, a simple search over all possible $\rho$ would suffice for identifying candidate experiments. The case where experiment designs vary in terms of $\tau$ is more complex; optimal $\tau$ for an experiment would be dependent on unknown effects $\theta_x$; selecting a good candidate experiment would involve estimating the optimal $\tau$ given posteriors over $\theta_x$. This would be an interesting problem to consider on its own as a future research direction.
>
> We now discuss in a **new Appendix E**, the seemingly limited scope of Bayes-OCP in terms of the design dimensions it can adjust adaptively, and how the high-level strategy of Bayes-OCP should actually be applicable to any set of viable experiment designs, including ones with varying $\tau$ and $\rho$, with suitable adjustments to the process of identifying candidate experiments.

---

### Official Review · Reviewer_jmHo · 2022-11-19

**Confidence:** 3
**Correctness:** 4
**Technical Novelty And Significance:** 3
**Empirical Novelty And Significance:** 3
**Recommendation:** 8

**Clarity, Quality, Novelty And Reproducibility:**

The paper is well-written and easy to follow. I also don't have real concerns about the quality, novelty and reproducibility of this paper. However, there are also some weaknesses as detailed in the previous section of the review. And also I have the following questions and suggestions for the authors.
* Is it correct that throughout the paper, the authors are considering targeting a single/unique subpopulation within $\mathcal{X}$? Can this be generalized to multiple targeting subpopulations? This might be helpful to include A/B testing, which is a very important experiment design example, as a special case. Or can A/B testing already be modeled with a single targeting subpopulation?
* Why isn't the adaptive signature design mentioned in Table 2 compared in the experiments?
* When computing the average metrics (e.g., utility) over multiple instances, some normalization might be needed to make it a fairer comparison (since otherwise an instance with a very large metric/utility will dominate).
* As mentioned above, Figures 3 and 4 need more explanations and many details of the figures are not easy to understand. For example, in Figure 3, why is it the case that only greedy Bayes-OCP (but not the non-greedy Bayes-OCP) is compared in the top figure? Why is futility stopping not compared? Are the green, amber and red curves all for greedy Bayes-OCP? And is the top figure an amber instance and the bottom one a red instance? Just to give some example questions the readers may have so that the authors can explain in the paper to improve the readability.
* Just to check, the main setting of this paper is online decision making, but the discretized RL algorithm in Appendix A needs to know the full model, right? This is not an issue but just want to clarify.
* In Appendix B.1, it would be nice if the authors can provide a detailed algorithm frame for the Bonferroni correction variant for better reproducibility.



**Strength And Weaknesses:**

Strengths:
* The framework is well motivated and the applicability of the model to different domains like healthcare, finance and energy is clearly stated and demonstrated.
* The theoretical results of the special case OCP solution characterization are inspiring. The numerical results are also generally convincing and supportive.
* The appendix also contains many interesting discussions, including the POMDP formulation of OCP, RL benchmark, the Bonferroni correction to improve FWER and the robustness to outcome model mis-specifications.

Weaknesses:
* If I understand correctly, it is assumed that the $R_{\psi}$ and $C_{\psi}$ coefficients in the utility function are known a priori for each experiment design $\psi$, which again limits the applicability and practicality of the proposed framework. Also some explanations are needed to characterize when such an assumption holds and when not.
* The figures (Figures 3 and 4) in the numerical experiments are not very easy to follow (the tables are good and clear, though). Some more explanations on them (especially how the switching points are determined, and how to associate each algorithm to each of the curves, etc.) would be very helpful.
* The discretized benchmark for solving POMDP formulation is not a very fair comparison. There are lots of more efficient POMDP algorithms (both with and without fully known models) that can be used (and many off-the-shelf solvers are available such as https://github.com/JuliaPOMDP/POMDPs.jl), and the authors should better compare with them in order to argue that the POMDP + RL approach does not work (or not).

**Summary Of The Paper:**

The authors formulate the optimal commitment problems (OCP), which can be used to model problems in which the agent determines whether to continue with some long-term actions like experiments until a signal of success or failure is received or to stop it in the middle. The agent is also allowed to change to a different (long-term) action during the process. On the theoretical side, the authors characterizes some fundamental properties of the optimal solution in a simplified OCP problem that has a single commitment. On the empirical side, the authors then propose an algorithm, Bayes-OCP that can be used to solve more general OCP problems. The authors also conduct  experiments to demonstrate the advantage of the proposed model and algorithm over existing models/algorithms in the literature.

**Summary Of The Review:**

Overall I think the paper contains non-trivial contribution in terms of both modeling, theory and practice. It's also clearly written and the reproducibility is also good given the detailed experiments. But the authors should also take a further look at the potential weaknesses detailed above and try to clarify and/or fix them.

---

> ### Author Response · Authors · 2022-11-22
> **Response to Reviewer jmHo [Part 4/4]**
>
> **Answers to Questions Regarding Clarity**
>
> * We consider targeting only one population per experiment run by Bayes-OCP but this population can consist of any combination of atomic-populations $x\in\mathcal{X}$. Throughout the paper, we use $\mathcal{X}$ to denote the set of atomic-populations, $x\in\mathcal{X}$ to denote an individual atomic-population, and $X\subseteq\mathcal{X}$ to denote a collection of atomic-populations, which we simply call a population.
>
> * Adaptive signature designs are only applicable when $\mathcal{X}$ is a continuous set. The equivalent benchmark for discrete sets is an adaptive enrichment design, which we consider in our experiments. Unfortunately, our algorithm is not applicable to cases where $\mathcal{X}$ is continuous, and hence, cannot be compared directly with an adaptive signature design.
>
> * For the objective we desire to optimize, a normalization procedure is not only not needed but also should be avoided. Given a prior distribution over environment instances, we want to maximize the expected utility. If there is an instance with a severely negative utility, even when that instance is relatively unlikely, better methods should be robust against it to keep its average utility high. In such a case, normalizing utilities achieved in individual instances, would favor methods that can achieve higher utilities in a majority of instances but fail catastrophically for some.
>
> * See our response above for “Figures 3 and 4;” we will discuss these questions in the new appendix we plan to add that will break down Figures 3 and 4 into their individual components.
>
> * Yes, indeed the RL algorithm in Appendix A is unique among our other benchmarks in the sense that it requires a training phase with a given model of the environment. However, it should be noted that this model does not contain any more information than what would be available for our other benchmarks. Only drawback of a training phase is that, for different environment parameters such as $R_{\psi}$ and $C_{\psi}$, an optimal policy would need to be re-trained whereas Bayes-OCP can be applied straightaway.
>
> * We will include modified algorithms with Bonferroni correction to Appendix J (in the revised version). Although we would like to highlight that the changes are very minimal: We cap the number of total experiments $n$ at two and set $\alpha=F^{-1}(0.975)$ instead of $\alpha=F^{-1}(0.95)$.

---

> ### Author Response · Authors · 2022-11-22
> **Response to Reviewer jmHo [Part 3/4]**
>
> **The RL-based Benchmark**
>
> In the **revised Appendix A.1**, we now compare Bayes-OCP against a more sophisticated RL-based benchmark. Specifically, we follow the approach of a recent paper (Ni et al. “Recurrent model-free RL can be a strong baseline for many POMDPs,” ICML, 2022) for solving complex classes of POMDPs, which the optimal commitment problem is one of as we have already highlighted in Appendix A. The results are given in the table below (see Table 4 in the revised paper), we see that Bayes-OCP performs better.
>
> | Algorithms: | Futility Stopping w/ Ni et al. 2022 | Futility Stopping w/ Bayes-OCP |
> |-------------|:------------------------------------:|:------------------------------:|
> | Utility     | 78.8                                 | **150.0**                      |
> | FWER        | 0.0%                                 | 0.1%                           |
> | Switches    | 0.7                                  | 0.5                            |
> | Success     | 24.2%                                | 45.4%                          |
> | T-to-S      | 600.0                                | 600.0                          |
> | T-to-F      | 23.6                                 | 57.6                           |

---

> ### Author Response · Authors · 2022-11-22
> **Response to Reviewer jmHo [Part 2/4]**
>
> **Figures 3 and 4**
>
> We agree with the reviewer that Figures 3 and 4 are information dense; this is due to the limited space available. In the camera-ready version of the paper, we will plot Figures 3 and 4 individually for each benchmark *with only the curves relevant to that benchmark* in an appendix. This should hopefully make it clear how switching/stopping points are determined for a given algorithm, and which curves play a role in determining those points.

---

> ### Author Response · Authors · 2022-11-22
> **Response to Reviewer jmHo [Part 1/4]**
>
> Thank you for your thoughtful comments and suggestions! Please find our answers as follows:
>
> **Rewards and Costs**
>
> Indeed, we assume $R_{\psi}$ and $C_{\psi}$ to be known, which have now been clarified in the revised version of our paper. In the context of clinical trials, which have been our main focus in terms of application, $C_{\psi}$ would correspond to the estimated cost of operating the experiment $\psi$; this would need to be determined according to the historical costs of running experiments and would depend on factors like how easy it is to recruit patients from the desired target population $X$. $R_{\psi}$ would correspond to the projected revenue of marketing an approved drug to patients; this too would primarily depend on the size of population which the treatment is approved for and hence can be marketed to. While assuming $R_{\psi}$ and $C_{\psi}$ to be known (or estimated/projected well) limits applicability, the main factor driving these values—that is the target population $X$ of an experiment $\psi$—is always well characterized (it needs to be well characterized to be able to run the experiment in the first place). Since the unknown treatment effect remains as the main source of uncertainty during the development journey of a new treatment—specifically, in whether a new treatment will be approved or not—we have chosen to focus on the case with unknown treatment effects only.
>
> We will include this discussion in the camera-ready version of the paper. Unfortunately, at this point of the review process, it is no longer possible to upload another revised version.

---

### Decision · Program_Chairs · 2023-01-20

**Decision:**

Accept: poster

**Justification For Why Not Higher Score:**

Results are a bit contrived

**Justification For Why Not Lower Score:**

The weird setting of bandits/online learning is nice (though the literature survey misses many relevant references)

**Metareview: Summary, Strengths And Weaknesses:**

This paper looks at some optimal stopping problem where the objective is to stop a task and start a new one if the former is not "successful" (as starting new task has some cost).

If there is one task, the problem is not that difficult, but it gives insight to the cases of multiple tasks, where the analysis is more empirical than theoretical, which is a slight concern (I believe the latter is feasible and would provide more insights than the mere empirical one).

Anyway, the reviewers are rather positive, and I find this paper interesting as well (the fact of having non-standard repeated problems, i.e., interesting new potential variants of bandits is quite nice). Hence I recommend acceptance

**Note From Pc:**

if the above contains the word "oral" or "spotlight" please see: "oral" presentation means -> notable-top-5% and "spotlight" means -> notable-top-25%. As stated in our emails, we are disassociating presentation type from AC recommendations